# Neural and behavioral adaptations to frontal theta neurofeedback training: A proof of concept study

**Scott E. Kerick**[1]*, **Justin Asbee**[2], **Derek P. Spangler**[1,3], **Justin B. Brooks**[1,4,5], **Javier O. Garcia**[1], **Thomas D. Parsons**[6], **Nilanjan Bannerjee**[5], **Ryan Robucci**[5]

**1** U.S. Combat Capabilities Development Command, Army Research Laboratory, Aberdeen Proving Ground, Aberdeen, MD, United States of America, **2** The Institute for Integrative & Innovative Research, University of Arkansas, Fayetteville, AR, United States of America, **3** Department of Biobehavioral Health, Penn State University, University Park, PA, United States of America, **4** D-Prime, Washington, DC, United States of America, **5** Department of Computer Science and Electrical Engineering, University of Maryland at Baltimore County, Baltimore, MD, United States of America, **6** Computational Neuropsychology and Simulation (CNS) Laboratory, Edson College, Arizona State University, Phoenix, AZ, United States of America

* scott.e.kerick.civ@mail.mil

**Data Availability Statement:** The data underlying the results presented in the study are available from Open Science Framework (https://osf.io/

## Abstract

Previous neurofeedback research has shown training-related frontal theta increases and performance improvements on some executive tasks in real feedback versus sham control groups. However, typical sham control groups receive false or non-contingent feedback, making it difficult to know whether observed differences between groups are associated with accurate contingent feedback or other cognitive mechanisms (motivation, control strategies, attentional engagement, fatigue, etc.). To address this question, we investigated differences between two frontal theta training groups, each receiving accurate contingent feedback, but with different top-down goals: (1) increase and (2) alternate increase/ decrease. We hypothesized that the increase group would exhibit greater increases in frontal theta compared to the alternate group, which would exhibit lower frontal theta during down- versus up-modulation blocks over sessions. We also hypothesized that the alternate group would exhibit greater performance improvements on a Go-NoGo shooting task requiring alterations in behavioral activation and inhibition, as the alternate group would be trained with greater task specificity, suggesting that receiving accurate contingent feedback may be the more salient learning mechanism underlying frontal theta neurofeedback training gains. Thirty young healthy volunteers were randomly assigned to increase or alternate groups. Training consisted of an orientation session, five neurofeedback training sessions (six blocks of six 30-s trials of FCz theta modulation (4–7 Hz) separated by 10-s rest intervals), and six Go-NoGo testing sessions (four blocks of 90 trials in both Low and High time-stress conditions). Multilevel modeling revealed greater frontal theta increases in the alternate group over training sessions. Further, Go-NoGo task performance increased at a greater rate in the increase group (accuracy and reaction time, but not commission errors). Overall, these results reject our hypotheses and suggest that changes in frontal theta and performance outcomes were not explained by reinforcement learning afforded by accurate

3eqdh/). POC: Scott E. Kerick (scott.e.kerick.
civ@army.mil).

**Funding:** JA: Research was sponsored by the
Army Research Laboratory and was accomplished
under Cooperative Agreement Number W911NF-
21-2-0097. The views and conclusions contained
in this document are those of the authors and
should not be interpreted as representing the
official policies, either expressed or implied, of the
Army Research Laboratory or the U.S.
Government. The U.S. Government is authorized to
reproduce and distribute reprints for Government
purposes notwithstanding any copyright notation
herein. NB, RR: Research was sponsored by the
Army Research Laboratory and was accomplished
under Cooperative Agreement Number W911NF-
19-2-0106. The views and conclusions contained
in this document are those of the authors and
should not be interpreted as representing the
official policies, either expressed or implied, of the
Army Research Laboratory or the U.S.
Government. The U.S. Government is authorized to
reproduce and distribute reprints for Government
purposes notwithstanding any copyright notation
herein. The funders had no role in study design,
data collection and analysis, decision to publish, or
preparation of the manuscript.

**Competing interests:** The authors have declared
that no competing interests exist.

contingent feedback. We discuss our findings in terms of alternative conceptual and meth-
odological considerations, as well as limitations of this research.

## Introduction

Neurofeedback (NF) is "a non-invasive brain stimulation technique equipped with a closed-
loop control mechanism, whereby information on the dynamics, usually non-observable, is
made observable to subjects, who can then use it to retroact on it, and push it towards func-
tionally desirable goal states. NF involves defining: i) the general goal; ii) a neural target as fea-
ture; iii) an appropriate stimulation schedule" [1] (p. 2). NF training is based on operant
conditioning principles [2], and involves both implicit (automatic) and explicit (volitional)
learning mechanisms [3]. Depending on one's goals, training can target various brain regions,
frequency bands, and functional outcomes. Scientific evidence of a clear relationship between
the particular neural feature targeted by NF training and the behavioral or cognitive function
being studied is measured by unambiguous changes in the targeted neural feature and by
changes in the targeted behavioral or cognitive function [4]. In the present study, the goal is to
induce changes in frontal theta power to produce changes in executive task performance.

Many NF paradigms target the modulation of neural functions related to executive control:
the higher-order brain processes that support goal formation, planning, monitoring, and con-
trolling complex, goal-directed thoughts and behaviors. Central to this system are the anterior
cingulate cortex (ACC), which monitors for conflict and the demand for control, and the pre-
frontal cortex (PFC), which exerts goal-directed modulatory influences on sensory, perceptual,
and memory processes represented in subcortical and posterior-cortical regions [5–9]. High
levels of executive control are crucial for optimal human performance on many tasks ranging
from the mundane (e.g. social interaction, writing a paper) to high-pressure activities where
errors can have lethal consequences (e.g., military battlefield operations, air traffic control). As
such, using NF to train modulation of neural functions related to executive control is a promis-
ing noninvasive means to potentially shape human performance and thus enhance perfor-
mance, safety and well-being.

### Theta: A promising neural target for training executive control

Extensive research has shown that EEG-theta synchronization is central to executive control
[10–16]. Theta synchronization functions to enable long-distance communication in the
brain, dynamically linking subcortical and cortical brain networks associated with integrating
and coordinating perceptual, cognitive, and motor control functions. Such theta dynamics are
biased by top-down goal states "to process and evaluate sensory inputs according to their val-
ues to achieve the goal. . . [and] govern how incoming sensory inputs map onto possible
responses" [16] (p. 3). Accordingly, frontal theta NF training for enhancing executive cognitive
functions is of great scientific and applied interest.

In a meta-analytic review of the NF literature, Rogala et al. [4] concluded that, "the validity
of EEG-NFB protocols can be measured by unambiguous changes in EEG activity and by
changes in the targeted cognitive function. Unfortunately, most of the work conducted in the
EEG-NFB field has failed to satisfy the unambiguity criterion for both of the variables and the
field itself has shown a big tolerance for violations of scientific methodology" (p. 2). They went
on to say that "a rigorous scientific approach to EEG-NFB is rare, and experiments performed
on healthy participants to study the effectiveness and/or mechanism of training are very

limited" (p. 8). Based on their review, only five theta NF studies met these criteria [17–21]. Four of five studies implemented theta up-regulation, and all five revealed significant EEG and behavioral changes in NF vs. control groups (more recently see [22, 23]).

As cited by Rogala et al. [4], most frontal medial theta (Fmθ) NF studies designed with the goal to facilitate executive control performance in healthy participants have instructed participants to *increase* Fmθ power (4–7 Hz) for prolonged trial periods (i.e., to produce sustained or tonic increases over several minutes). Overall, this research reveals that young adults who exhibit greater increases in sustained Fmθ power over NF training sessions also exhibit greater improvements on some but not all executive tasks relative to those who receive non-contingent or sham (false) NF [19, 21, 23]. For example, in Enriquez-Geppert et al. [19, 24], participants were instructed to increase individual Fmθ activity (average of sites Fz, FC1, FCz, FC2, Cz), as long and saturated (highest power levels) as possible continuously over each of six 5-min blocks in each of eight sessions. To determine individual Fmθ frequency, they recorded event-related Fmθ power during a battery of four executive control tasks (three-back, task-switch, Stroop, and stop-signal tasks) in which each trial in each task was under 3 s. Peak phasic Fmθ was observed approximately 100–500 ms post-stimulus onset averaged across tasks. In both studies, participants in the NF group exhibited greater tonic increases in Fmθ power over sessions compared to those in the pseudo NF group (received feedback signals recorded from a matched participant in the real feedback group). In [24], post-training task performance was not evaluated, but in [19] performance was evaluated pre- and post- Fmθ training on the same battery of tasks. They found that the NF group exhibited higher post-test event-related (phasic) Fmθ power and performance improved pre-post training in both NF and pseudo NF groups, but to a greater extent in the NF group on two of the four tasks (three-back and task switch) vs. the pseudo NF group. The finding of performance increases in the pseudo NF group in three-back and task switch tasks suggests non-specific effects such as learning, attention, effort, etc. [22, 25]. It is also interesting to note that NF training designed to tonically increase Fmθ power was associated with phasic increases during post-test task performance. This finding suggests non-specific temporal brain dynamics between training (tonic) and outcome (phasic) states. Additionally, they also observed marginal trends for alpha and beta power increases across NF training sessions in the NF vs. pseudo NF group, suggesting lack of frequency specificity effects as well.

More recently, Brandmeyer & Delorme [23] also compared Fmθ NF training (Fz) vs. a sham control group (received feedback signals recorded from a matched participant in the real feedback group) and also instructed participants to tonically increase Fmθ activity (using fixed [4–6 Hz] rather than individualized theta bands) as long and saturated as possible continuously over each of six 5-min blocks in each of eight sessions while simultaneously monitoring breathing cycles. Executive task performance was evaluated pre- and post-training on N-back (1-, 2-, & 3-back), Sustained Attention to Response (SART), and Local-Global conflict tasks; each task also consisting of sequences of short trials (<3 s). Consistent with Enriquez-Geppert et al. [19, 24], participants in the NF group exhibited greater tonic increases in Fmθ power over sessions compared to those in the sham group. Also consistent with [19], some, but not all, behavioral outcomes improved significantly more in NF vs. sham group subjects. Specifically, the NF group demonstrated faster reaction times (but not accuracy) for correct trials in the N-back task, but no differences were observed for the SART or Local-Global conflict tasks. These findings also suggest non-specific temporal brain dynamics between training (tonic) and task (phasic). Additionally, they also observed alpha and beta power increases in the NF, but not sham, group across training sessions, suggesting lack of frequency specificity effects as well. Consistent with Enriquez-Geppert et al. [24], Brandmeyer & Delorme [23] observed that Fmθ NF training was associated with improvement on the N-back memory task but not

sustained attention or conflict monitoring tasks, indicating some level of outcome specificity across studies (working memory).

Wang & Hsieh [21] investigated the effects of Fmθ (4–7 Hz at Fz) up-regulation NF training vs. sham control groups (twelve 15-min sessions) in elderly vs. young participants on attention (Attention Network Test; ANT) and working memory (Sternberg's recognition memory task) performance. Sham control in this study consisted of participants modulating randomly selected frequency bands from 10–13, 13–16, 16–20, or 20–25 Hz for each session. They found that both younger and older NF training groups exhibited increased Fmθ over sessions, and that both NF training groups (but not sham groups) improved on the orienting network task, and the older (but not younger) NF group improved on the conflict network and Sternberg recognition tasks, whereas sham groups did not. Here again, memory function was improved with Fmθ NF training, but in contrast to Enriquez-Geppert et al. [24] and Brandmeyer & Delorme [23], conflict monitoring and attentional orienting functions also improved, suggesting generalizability of Fmθ NF training.

We are not aware of NF research designed to promote *decreases* in Fmθ in healthy young adults to enhance executive task performance. However, in aging and clinical populations, where abnormally high levels of theta are associated with cognitive impairments, NF training to decrease theta power has been shown to be associated with improvements in cognitive functions [17, 22, 26–28]. Often these studies also involve simultaneous modulations of other frequency bands (e.g., alpha, sensorimotor rhythm, beta) and brain regions (e.g., central, parietal) so the specific role of Fmθ in NF training is not always clear. In a study designed to down-regulate abnormally high theta activity in elderly (thirty 30-min sessions), over individual-specific regions where it was empirically highest in elderly patients relative to database norms, Becerra et al. [17] found that, compared to a control group receiving sham (random) feedback, the down-regulation feedback group exhibited decreased theta power and greater performance improvements in verbal comprehension, verbal IQ, and memory. Neither group improved performance on attention or executive control tasks, while the control group actually showed a deterioration in executive task performance with sham training. However, these findings were observed for training of various electrode locations and cannot be attributed to Fmθ NF training. Further, it should be noted that NF training goals for clinically healthy older adults with abnormally high levels of theta activity are to prevent cognitive deterioration more so than to improve cognitive performance as is the goal for healthy participants.

Overall, research on Fmθ NF training has revealed that young adult participants instructed to sustain elevated levels of Fmθ during training is associated with phasic theta power increases and improvement on some, but not all executive tasks relative to those who receive sham NF [19, 21, 23]. Regarding temporal dynamics, we're aware of no studies that have investigated shorter duration training trials to elicit more phasic adaptations in a more task-specific manner to the targeted outcomes. Similarly, no studies have compared up- vs. alternating up- and down-regulation training that we're aware (but see [29] cf [30]). Both of these factors may relate to questions about outcome specificity with respect to executive tasks in which excitation and inhibition processes are dynamically changing over short time scales (ms-sec).

## Unresolved issues and gaps

One issue of previous NF training studies which has not been fully addressed concerns the sham control. Sham control conditions have typically been implemented as various forms of false feedback (e.g., receiving feedback from another participant's previous recordings) or random feedback from one's own recordings (e.g., non-contingent reward). Considering that accurate contingent feedback linking response and reward is required for optimal learning to

occur [31], it's assumed that participants receiving real NF training have learned to modulate specific EEG features (e.g., Fmθ), while sham participants have not. However, it cannot be ruled out from previous research whether differences in Fmθ between real and sham groups are due to self-regulation based on contingent reward learning mechanisms. Alternative mechanisms are also plausible, e.g., increased frontal theta has also been shown to be associated with attentional effort/task demand [11, 32] (Gevins et al., 1997; Rajan et al., 2018), decision making [16], working memory [33], and boredom/fatigue [34, 35], to list a few. Thus, it's conceivable that differences in Fmθ may also be at least partially attributable to other cognitive functions, especially considering that sham control participants have been shown to report knowledge of lack of contingency between their strategies and feedback signals [23], which may reduce motivation and attentional engagement. It's also common that subsets of NF training participants are non-responders [23, 24], revealing that even with accurate contingent feedback some participants are unable to increase Fmθ. Further, sham groups have also shown increases in pre-post task performance [19], suggesting that behavioral changes may be related to some non-specific factors unrelated to contingent reward mechanisms of NF training. Contingent reward enables the potential for accurate coupling of top-down goals and bottom-up stimulus processing (error monitoring and control) that may account for the learning differences observed in previous sham control studies. However, a need exists to compare NF training protocols in which accurate contingent reward is held constant, while manipulating other parameters of training, such as the top-down training goals (e.g., increase versus decrease Fmθ).

Another key limitation of previous NF studies, including those targeting Fmθ, is that they modulate neural activity only in one direction (either solely up-modulation or solely down-modulation). However, to support adaptive behavior in real-world situations, neural activity must exhibit dynamic variability such that activation and inhibition must be coordinated in response to varied internal/external demands. There are a number of theoretical concepts rooted in nonlinear systems that underscore the importance of such dynamic variability such as *self-organized criticality* [36–40], *multistability* [41–43], and *coordination dynamics* [38, 42]. Here, a system (e.g., the brain) with high levels of dynamic variability is on the verge of instability, such that this system is more flexible (i.e., neural resources can be more efficiently adapted to changing internal and external demands). In these views, it is not sustained increases or decreases in theta (or any other neural process) that are most optimal for adaptive cognitive performance. Rather, dynamically shifting between increasing and decreasing activity (up and down modulation) might promote more efficient neural adaptations in support of superior executive control performance. Previous NF research consists of training sustained Fmθ activation (5–30 min continuous), whereas in the performance of executive tasks, frontal theta phasically increases and decreases in an event-related manner (ms-sec) over trials. In this regard, we agree with Papo [1] that NF training should consider dynamics-to-function mapping. However, previous research suggests that tonic increases in Fmθ over training sessions translates to phasic increases in Fmθ during performance of some executive tasks pre-post training [19]. Therefore, further research is needed to investigate temporal dynamics and power modulations in NF training.

Representing another key issue, analogous to the above on sham control issues, the Fmθ activation targeted by NF is often confounded with the neural functions required to modulate brain responses in NF protocols. Specifically, NF is an attention-demanding task requiring the activation of frontal executive control networks during training irrespective of the frequency or region targeted by feedback [25, 44, 45]. For example, Gruzelier [25] contended that "evidence that a frontal locus is the strongest locus showing changes following training, though this is not an exclusive locus. Even posterior training predominantly leads to stronger frontal than posterior effects" (p. 167) and that Fmθ "may represent a nexus of factors including

learning, self-regulation, as well as non-specific issues such as attention and effort, but perhaps most central of all action monitoring" (p. 171). Because Fmθ is implicated in such a nexus of executive control processes, observed Fmθ increases during NF training may be erroneously attributed to effective NF training (i.e self-regulation based on accurate contingent feedback) when such increases may merely indicate sustained deployment of attention, effort, and action monitoring. Although sham control theoretically controls for attention demands and cognitive effort while precluding reinforcement learning afforded by accurate contingent feedback, evidence suggests that sham control participants may eventually discover that their strategies and cognitive effort to control the feedback signal are not reliable [23], and thus motivation, cognitive effort, and attention engagement may wane over the course of training. This possibility obscures extant findings on Fmθ modulation, such that observed effects of Fmθ modulation on the brain and performance may not be driven by NF *per se* (i.e. self-regulation) but instead by differences in theta activation related to attention engagement or other non-specific executive control processes when attempting NF. In order to test whether training-related increases in Fmθ are indeed associated with self-regulation based on contingent reward learning mechanisms researchers could examine whether participants attempting alternating periods of up/down-modulation exhibit higher/lower Fmθ during training relative to a comparison group which only attempts Fmθ up-modulation. Here, up/down modulation would require alternating top-down NF goals. As such, if up/down-modulation promotes lower Fmθ levels, particularly in blocks of down- versus up-modulation within the alternating group, and relative to the group trained in up-modulation alone, then NF training adaptations are likely due to self-regulation based on accurate contingent feedback. If, on the other hand, up/down modulation promotes higher Fmθ levels relative to up-modulation alone, then NF training adaptations are likely due to some other mechanism unrelated to self-regulation. In other words, if observed Fmθ increases are in fact due to NF training effects (self-regulation), then up-modulation should promote greater Fmθ than alternating up and down modulation. Further, if training-related Fmθ power increases are associated with improvements in executive task performance, then up-modulation training should lead to superior executive task performance.

## Current study

From a dynamical systems perspective, it is conceivable that designing NF to more flexibly train variable multistable states (e.g., transitioning between activation and inhibition) by rapidly alternating increasing and decreasing Fmθ feedback may lead to improved executive task performance relative to feedback designed to more rigidly train monostable states (e.g., consistently high or low activation). We are not aware of any NF studies which have asked this question, however, Papo [1] suggested that future NF studies should be designed with the principle of dynamics-to-function mapping in mind (i.e., neural dynamics training with greater outcome specificity). Accordingly, in the present study, we investigated differences in neural and behavioral adaptations to NF training between two groups of participants receiving different training protocols: (1) Alternating up/down-modulation of Fmθ (ALT), and (2) Increasing or up-modulation of Fmθ (INC). We investigated whether up-down modulation (ALT) of Fmθ led to superior improvements in executive control performance of a Go-NoGo task, which requires dynamically changing behavioral activation and inhibition, relative to just up-modulation (INC; common method employed by prior studies).

In both NF training groups, accurate contingent feedback was provided and short discrete trials (30-s) followed by a brief rest (10-s) were implemented rather than long sustained trial periods (several min). We implemented such short trials in an effort to promote dynamic variability in brain state that purportedly supports the dynamic executive control functions

targeted by the NF training (i.e., dynamically alternating between goal states of activation and inhibition in the Go-NoGo task). We were also interested in whether Fmθ up-modulation training was truly driven by effective NF (self-regulation based on accurate contingent reward), as inferred in previous research, or by differences in non-specific executive control processes deployed during NF training (mental effort, attentional focus). This question was addressed with our aforementioned design where we compared Fmθ responses between alternating up/down-modulation to up-modulation alone. We tested two competing hypotheses regarding the effects of NF training on concomitant levels of Fmθ power. If Fmθ changes are due to self-regulation afforded by accurate contingent feedback, then the INC group should exhibit greater increases within and across training sessions. If, on the other hand Fmθ increases more in the ALT group, or no differences are observed between groups, then self-regulation based on accurate contingent reward mechanisms underlying reinforcement learning would be challenged. These hypotheses assume that self-regulation processes based on operant conditioning are differentiable from non-specific executive control processes. Further, they assume an adequate stimulation schedule to enable sufficient reinforcement learning. Assuming that Fmθ is effectively shaped by self-regulation in both groups, we hypothesized that the INC group would exhibit greater increases in Fmθ within and across NF training sessions. However, in contrast with previous research, we hypothesized that the ALT group would improve at a greater rate on the Go-NoGo shooting task over sessions, implying that training to dynamically modulate increases and decreases in Fmθ provides greater training specificity of dynamics-to-function mapping than does training to systematically increase Fmθ. Furthermore, we hypothesized higher Fmθ power in the ALT group during performance of the Go-NoGo task over sessions, suggesting more refined task specificity and context-dependent adaptations, or more specific dynamics-to-function mapping, in the ALT group.

## Materials and methods

### Participants

Thirty (N = 30; 13 female) young healthy adults (ages 18–40 yr; mean 24.99 ±3.21) participated in seven separate sessions within a three-week interval: one Orientation Session, five combined NF training + Go-NoGo shooting (SH) task testing sessions, and one post-training SH task testing session. Electroencephalography (EEG) was not recorded in the Orientation Session but was continuously recorded during NF training and SH task periods throughout each of the following six sessions.

### Study design and timeline

The participants were randomly assigned to either the INC or ALT group. Those in the INC group (n = 12; 5 female) were instructed to increase their Fmθ power during each of six trials in each of six blocks in each of five sessions. Those in the ALT group (n = 18; 8 female) were instructed to either increase or decrease their Fmθ during each of six trials within each block, with three blocks of increase and three blocks of decrease alternated by random assignment across the six blocks in each of five sessions. In the first session, the SH task was conducted before NF training as a pre-test performance assessment. In sessions 2–5, NF training was conducted before the SH task, and in the 6th session, only the SH task was conducted as a post-test performance assessment. Thus, participants in each group were administered five total NF training sessions and six total SH task test sessions. During the SH task, the participants performed Low and High and time-stress conditions in a counterbalanced order between participants, and within participants across sessions (see Fig 1). To investigate differences in Fmθ changes between Groups during NF training within and across sessions, we applied a Group

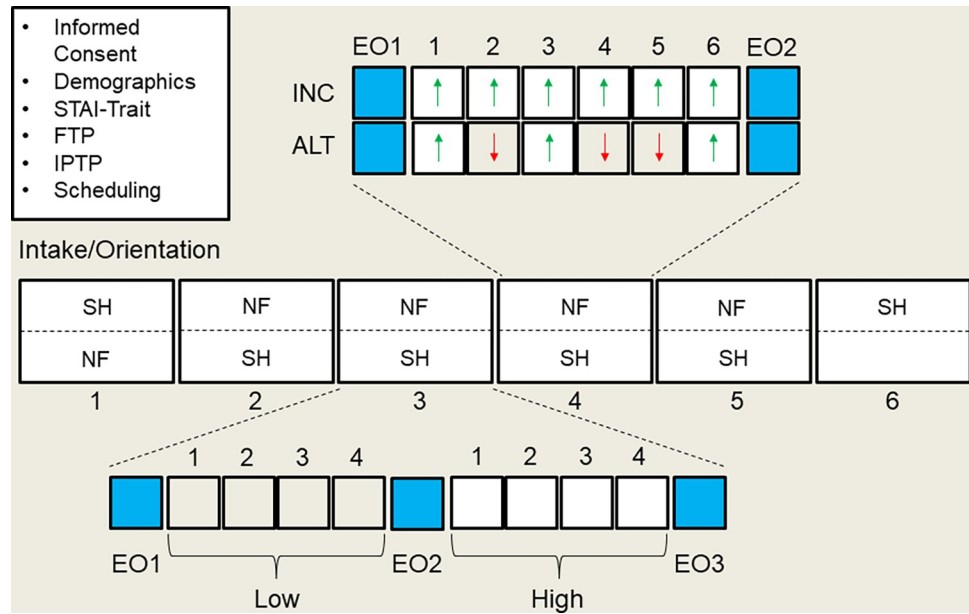

**Fig 1. Task design.** Experiment design illustrating the Intake/Orientation session, five sessions of NF training and six sessions of Go-NoGo SH task testing. Insert above shows six blocks (6 trials/block) of NF training for the INC (all increase Fmθ) and ALT (3 increase and 3 decrease Fmθ randomly assigned) groups with eyes-open rest periods before and after NF training (EO1-EO2). Insert below shows four blocks (90 trials/block) in both Low and High time stress conditions (counterbalanced order by subjects and sessions), with eyes-open rest periods before and after each SH task condition (EO1-EO3).

(ALT, INC) x Modulation (Up, Down) x Block (1–6) x Session (1–5) hierarchical mixed linear model. To investigate differences in Fmθ and behavioral performance changes between Groups during the SH task in Low and High time-stress conditions over sessions, we applied a Group (ALT, INC) x Session (1–6) x Condition (Low, High) hierarchical mixed linear model.

## Equipment and apparati

**Neurofeedback training and Go-NoGo shooting task.**  An HTC Vive virtual reality system (https://www.vive.com/us/) was used to generate and implement the NF training and simulated SH task testing environments and log events and behavioral responses of the participants in all NF training and SH task implementations. The Vive system consisted of a headset worn by participants to graphically display the immersive 3D environment and a wireless hand controller that features 24 sensors for unobstructed movement, and a base station that enables 360˚ motion tracking of the headset and controller in virtual space (120 Hz). A Tobii Pro X3-120 wearable eye tracking system (www.tobiipro.com) was integrated into the goggles to record pupillary and eye movement data. The headset was worn over the top of the EEG cap, which we verified through preliminary testing minimally interfered with either EEG signal acquisition or perception of the immersive environment. The hand controller (not used in NF training) was represented in the SH simulation environment as a semi-automatic pistol and the environment as a target range.

The SH task paradigm was programmed to simulate a target range from a first-person (egocentric) perspective and transmitted the following event markers online in real-time from the scenario generation computer via parallel cable connection to the EEG recording system: (1) onset of target exposure, (2), location of target (left, center, right), (3) distance of target (near, mid, far), (4) identity of target (enemy or friendly), (5) onset of trigger pull (if shot fired), (6)

result of shot (hit or miss target), and (7) offset of target exposure (time of target down if not fired upon or fired upon and missed).

**Electroencephalography (EEG) recording.** EEG data were acquired at 2048 Hz and referenced online to the Common Mode Sense (CMS) and Direct Right Leg (DRL) electrodes using a 64 (+8 external) channel BioSemi system (Amsterdam, The Netherlands; http://www.biosemi.com/products.htm). Continuous EEG recordings were acquired from 64 standard scalp locations (10–10 system [46] using BioSemi headcaps and a hypoallergenic, bacteriostatic gel was used to establish conductive electrode contact (NuPrep). Other non-invasive physiological measures (i.e., electrocardiography, blood pressure, photoplethysmography) were also simultaneously collected but are not of primary relevance to the current hypotheses. They are hence not described and/or reported in the current paper.

**Data integration and synchronization.** All data were integrated and synchronized using Lab Streaming Layer (LSL [47]; freely available at code.google.com/p/labstreaminglayer) and recorded continuously over each of two experimental periods separated by a 10–15 min break: (1) NF training including, pre- and post-training 60-s eyes-open (EO1, EO2) rest and (2) SH task, including Low and High time stress conditions and 60-s eyes-open rest preceding and following each shooting task condition (EO1-EO3). Thus, in each session, two separate raw continuous LSL files were generated (Extensible Data Format;.xdf), each of which consisted of rest and task data, as well as data recorded during intervals between rest and task intervals and between blocks of trials.

## Procedures

**Intake/Orientation session.** Interested participants were first asked to come to the Virtual Reality Laboratory at the Technology Research Center (TRC) at UMBC for an Intake/Orientation meeting. During this session, the participants were provided a more thorough description of the study and further explanation of procedures and demands, shown brief introductions and demonstrations of the NF and SH task simulations, physiological recording preparation and procedures, and encouraged to ask questions or express any concerns they might have before consenting to participate. Volunteers who agreed to participate were asked to read and sign an Informed Consent Agreement (approved by the Human Use Committee at ARL and the Institutional Review Board at UMBC, in accordance with the Declaration of Helsinki and the U.S. Code of Federal Regulations). At this time, the participant was randomly assigned to either the control or experimental group. To avoid potential biases or preconceptions of the participants in a manner that might confound the results of this study, we informed all participants that "The purpose of this study is to investigate how humans learn to control technological systems with their brain signals through short-term neurofeedback training". Thus, participants were blind with respect to the existence of control and experimental groups or the real purpose of the study. We then administered a basic demographics information form and the trait version of the State-Trait Anxiety Inventory (STAI [48]). These data were not analyzed in the present study. The participants were debriefed of the real purpose of the study following their completion of the study.

**Familiarization and training procedure (FTP).** To minimize novelty and learning effects during experimental testing, we first presented the participants a Familiarization and Training Procedure (FTP) during the Orientation Session. The FTP was conducted without physiological recordings or instrumentation beyond the Vive headset and hand controller, which was used in the implementation of the SH task. If the participant exhibited stable performance in the FTP (asymptotic hit rates on enemy targets), we proceeded to the individualized threshold testing procedure (described below), else we repeated the FTP until stable performance was

exhibited. In the FTP, a block consisting of 100 trials with all-enemy targets (presented with parameters: inter-target-intervals (ITIs) = 2000 ± 1000 ms spread over a Gaussian-distributed range of ITIs in 100 ms step increments presented in a randomized sequence and target exposure times (TETs) ranging from 400–1500 ms in equally distributed 100 ms bin step increments and presented in a randomized sequence across nine target positions. At this stage of training in the experiment, we sought to emphasize reflexive but accurate shooting responses to enemy targets to reinforce the buildup of a pre-potent response bias, which later in the experiment by design, would induce a greater likelihood of higher friendly-fire error rates during the experiment.

**Individual performance thresholding procedure (IPTP).** To induce a task-relevant time-stress effect, while accounting for individual differences in the ability to perform the task, we determined each participant's individual reaction time thresholds (i.e., individualized target exposure durations) corresponding to the 50th (High time stress) and 90th (Low time stress) percentile hit-rates for enemy targets using psychophysical methods (method of limits [49]) during the Orientation Session. In the method of limits, each participant was presented 10 trials at each of 11 TETs in alternating sequences of increasing and decreasing order. Individualizing TETs better ensured that the participants perceived the same relative time-stress effects during Low and High time stress conditions than would have by administering the same fixed target exposure times for all participants.

**Compensation and scheduling.** We provided monetary compensation to attract willing participants, reduce attrition, and incentivize completion of all training and testing sessions. We paid $10/hr as base pay for participation in each session and an additional $100 bonus for completion of the entire study. We scheduled dates and times of the six remaining sessions after the Orientation Session within a three-week interval (two sessions/week). In each training + testing session, the participants were instrumented for physiological recordings (~60 min) and asked to conduct NF training (~30 min) and perform the SH task (~45 min), except for the last session, in which they were asked only to perform the SH task.

**Neurofeedback training and Go-NoGo shooting task testing sessions.** *Neurofeedback training*. The Fmθ feedback signal was derived from FCz. FCz was referenced to averaged mastoids (A1/A2), bandpass filtered (0.5–30 Hz), and power spectral density was estimated over 1-s windows (2048 samples) with 875 ms overlap (1792 samples), and averaged over 4–7 Hz (8 estimates/s with 1 s history). Each session consisted of six blocks of six trials, and each trial consisted of a 10-s rest period followed by a 30-s active modulation period (180 s / block * 6 blocks = 1080 s / session [18 min] * 5 sessions = 5400 s total [90 min]). During the rest period, the participants were instructed to relax and fixate on the intersection of the origin (zero-point) of the XY axis of a blank graph. During the active modulation period, a line-plot beginning at the origin appeared and moved from left-to-right, updating in near-real-time reflecting increasing or decreasing power values relative to the current threshold, they attempted to increase or decrease the power values according to NF group instructions (see Fig 2). In the first session, pre-NF resting state theta power (4–7 Hz) at electrode FCz was defined as the threshold for NF power modulation throughout that session. In each subsequent session, the threshold for NF modulation was defined as the mean of FCz power across all blocks the previous NF training session. Artifacts were detected online based on amplitude thresholding (±75 μV) of FCz prior to spectral estimation. In the event of artifact detection, the feedback signal was displayed as a flat line at the last power level before artifact occurrence, and remained flat until the artifact subsided, after which ongoing power levels of the signal resumed.

All participants were instructed: "While attempting to control or modulate the feedback signal, use only your internal thoughts, feelings, mental imagery, attentional focus or concentration, or any mental process that may work for you. There is no right or wrong strategy or

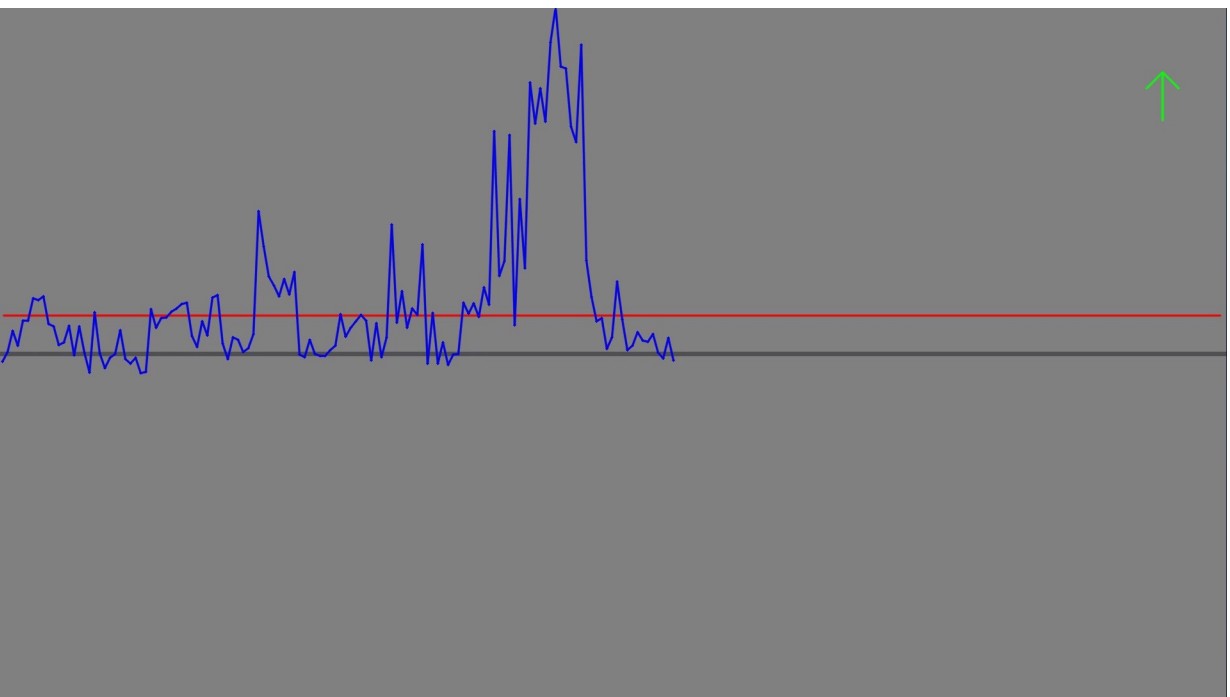

**Fig 2. Example NF training feedback.** Example 30-s single trial of NF modulation, partially completed. The black line is 0 dB, the red line is the goal threshold, the blue line is the feedback signal, and the green arrow is the up-regulation instruction indicator.

approach to modulating the signal, or your current brain state, as every individual will develop his or her own unique strategies or approaches that works for him or her. However, it's important that you do not intentionally generate noise or artifacts in the EEG recording to influence the signal amplitude, including intentionally moving or blinking your eyes, furrowing your brow, clenching your teeth, or tensing your facial, neck, or shoulder muscles or making gross postural movements by moving around in your chair".

For the ALT group, up- and down-modulation was randomly assigned across blocks, with the limitations that three blocks of each were implemented, and that no more than two consecutive blocks were the same. At the end of each block of trials, a pie chart was displayed at the bottom of the screen indicating the percentage of time participants' Fmθ power achieved the criterion threshold value. Thus, the participants were provided feedback in real time, as well as a summarization of cumulative feedback after each block of six trials. The participants were also asked to complete a set of five visual analog scales (VAS; 0-Not at all to 100-Extremely so) to assess subjective states of stress, fatigue, boredom, motivation, and attention. The five VAS scales were administered electronically via Graphical User Interface (GUI) immediately before and after NF training and before and after low and high time stress SH task periods in each session. These self-report data were unrecoverable due to technical issues and therefore could not be analyzed.

*Go-NoGo shooting task.* The Go-NoGo SH task in each of two time-stress conditions consisted of 360 pop-up targets pseudorandomly distributed 40 times at each of 9 range locations (three simulated distances (near, mid, far) by three lanes (left, center, right) and exposed at variable onset intervals (1000 ± 500 ms over a Gaussian-distributed range of 100 ms increments) for various target exposure durations. For each participant, the target exposure times determined in the ITPT were used to generate a distribution of 360 target exposure times separately for Low and High time stress conditions. The distributions of TETs were associated with

the pseudorandom distribution of targets to generate a target sequence file. Each session, the same sequence file (target type and location) was used for all participants (but with individualized TETs), and reshuffled into a different sequence each session. This procedure ensured that participants were given a different sequence of targets each session, but all participants received the same sequence in each session. In each time-stress condition, the task was composed of four blocks of 90 trials to minimize fatigue. At the end of each block, the simulation would pause, and the participants would take self-determined break durations up to 1-min. The total duration of SH task test sessions was ~45 min, including performance of the Low and High time-stress conditions, self-determined block breaks, and 1-min rest intervals preceding and following each SH task condition. The probability of targets (enemy) to non-targets (friendly) was .90/.10, respectively. This biased ratio favoring higher probability of target stimuli induces biased response tendencies priming the shooting response and challenging inhibitory control processes, thereby increasing the likelihood of friendly-fire errors. The participants were instructed to "shoot enemy targets as quickly and accurately as possible, while refraining from shooting at friendly targets".

The participants were also informed of a scoring system and feedback display in the form of a scoreboard at the back of the shooting range. The scoreboard was a digital display in which green reflects correct responses: shot fired at enemy targets (+1), hit enemy targets (+1), and inhibit shot-fired at friendly targets (+1); red reflects incorrect responses: shot-fired at friendly targets (-1; errors of commission), hit friendly targets (-1; compounded errors of commission), fail-to-fire at enemy targets (-1; errors of omission), and missed enemy target (-1; failed execution). Variable and jittered TET and ITI intervals were implemented in both conditions to prevent the participants from establishing a fixed rhythm of responding with constant stimulus durations and intervals between stimulus presentations, making the task more attention-demanding and reactive. Fig 3 is a screen capture of the SH Task.

*EEG processing and analyses.* EEG signal processing was applied using EEGLAB (ver 14.1.2b; http://sccn.ucsd.edu/eeglab/) and in-house code using MATLAB (9.3.0.713579 [R2017b]; Natick, MA) on a 64-bit Linux operating system. First, LSL data were imported, synchronized with other data streams (Biosemi, PortaPres, PPG, Vive events), and converted to Matlab format using either EEGLAB (loadxdf.m; https://github.com/sccn/xdf) or in-house Python scripts. Once in Matlab structure format, BioSemi data (including EEG, ECG, & respiration data) were extracted and partitioned by rest and task conditions based on event markers (separately for each NF and SH LSL file for each subject in each session).

*Pre-processing pipeline (PREP).* Following data partitioning, raw EEG data from each task and session were subjected to the Pre-Processing Pipeline (PREP [50]). None of the participants exhibited bad data requiring interpolation for FCz. The number of bad channels ranged from 0–20 (M = 5), and mostly consisted of edge electrodes in anterior frontal and lateral temporal electrodes due to poor contact or excessive muscle or 60 Hz artifacts in some subjects and sessions. The following pre-processing procedures were applied to the EEG data acquired in each experiment session: (1) highpass filtered (1 Hz) using a zero-phase finite impulse response filter (EEGLAB function pop_eegfiltnew); (2) down-sampled from 2048 to 512 Hz; (3) partitioned into rest and task files for each recording period ([NF: EO1, NF, EO2], [SH: EO1, SH(L), EO2, SH(H), EO3]); and (4) applied the Pre-processing Pipeline (PREP [50]) to data in each rest and task file separately (60 Hz and harmonics removed, bad channel detection and interpolation, and robust re-referencing to the average of all channels).

*Independent component analysis (ICA).* Following PREP, (5) the rest and task data files from each recording period (NF, SH) were separately re-concatenated (i.e., reverse of step 3); (6) epoched into non-event-locked, adjacent (non-overlapping), 1-s windows from the beginning to the end of the concatenated data; (7) applied automatic artifact-rejection algorithms to

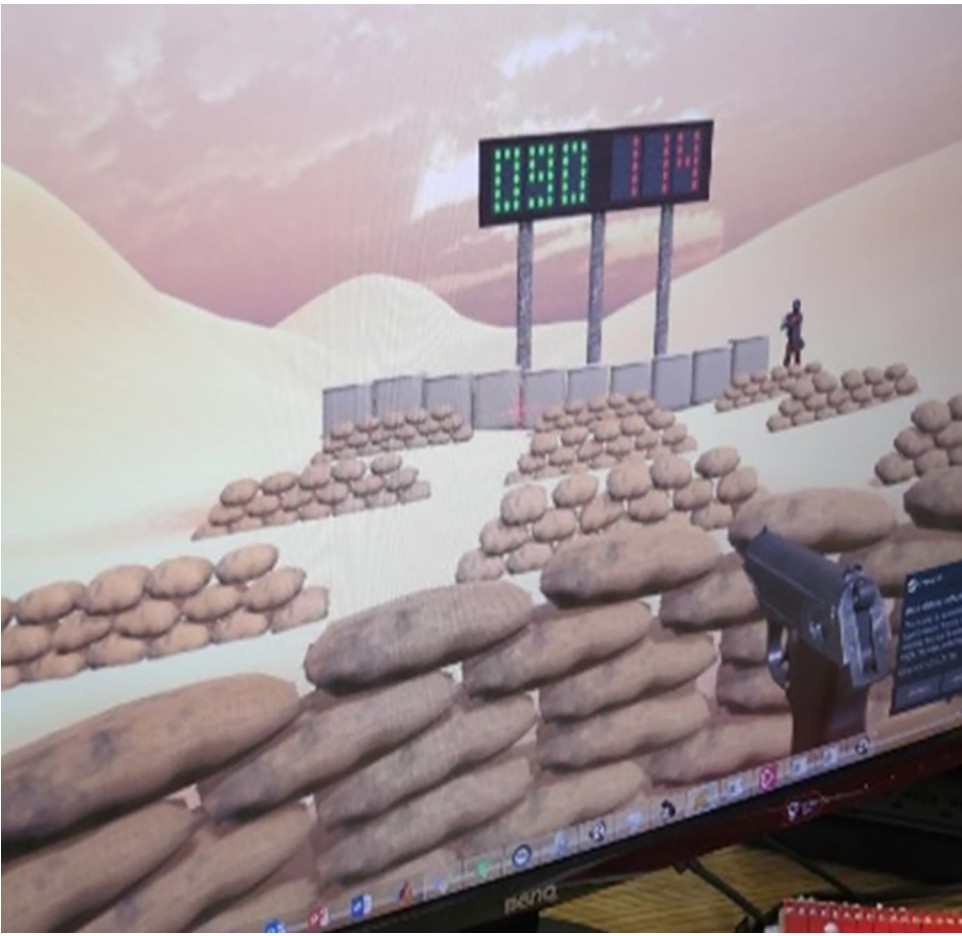

**Fig 3. Go-NoGo shooting task.** Single trial of SH task showing enemy target appearing at far right target location and current score feedback.

epoched data using EEGLAB function pop_rejmenu with the following rejection criteria: (a) amplitude threshold ($> \pm 100$ μV); (b) joint probability ($> 5$ SD); (c) abnormal trends (max slope 75 μV/1-s epoch; R-squared limit 0.3); and kurtosis ($> 5$ SD); (8) visually inspected all epochs and confirmed automatically-tagged epochs containing artifacts (or manually accepted or rejected epochs if incongruent with visual inspection); (9) deleted all artifact-tagged epochs; (10) applied independent component analysis (ICA [51]) and dipole source localization (dipfit) to the artifact-reduced, concatenated data; (11) selected and tagged ICs resembling eye-artifacts (based on IC time series, spectra, topographic maps, and dipole source locations; (12) copied ICA output (weights, inverse weights, sphere matrix, dipole source locations) from the concatenated data back to each independent data file; (13) rejected eye-artifact-tagged ICs in each data file; (14) re-visually-inspected each trial for residual artifacts.

*Spectral analyses*. For NF training, EEG data were epoched from– 3 to 30 s around the start of modulation for each trial. Data were then sub-epoched into 33 1-s non-overlapping windows for artifact detection and removal before spectral analyses. Pre-modulation baseline was defined as the period from -3 to 0 s and NF modulation from 0 to 30 s and spectral estimates were derived for each period separately using pwelch on 1-s windows with 50% overlap. For the SH task, EEG data were epoched from– 1 to 2 s around the onset of targets for each trial. Each trial was visually-inspected for artifacts and removed before spectral analyses. Pre-trial

baseline was defined as the period from -1 to 0 s and shooting from 0 to 2 s and spectral estimates were derived for each period separately using pwelch on 1-s windows with 50% overlap. Only task periods were analyzed in the present report (0–30 s for NF and 0–2 s for SH). The percentage of data retained for analyses was 112794 s / 154620 s * 100 = 72.95% for NF training and 120339 trials / 122153 trials * 100 = 98.51% for SH task.

*Behavioral performance analyses.* The following performance measures were extracted: percentage of trials indexing errors of commission (shot fired at friendly targets; % errors of commission), percentage of enemy targets hit (% hit; accuracy), and reaction times to enemy targets (ms; RTs). Participants were allowed to fire as many shots as they desired at each target. If more than one shot was fired at a target, target hits were registered if any of the shots hit the target. If a target was hit, RTs were computed as the time from target onset to the time of trigger pull of the shot that hit the target. If a target was missed, RTs were computed as the time from target onset to the time of the first trigger pull.

**Statistical analysis.** Multilevel linear modeling (MLM) was applied to examine effects of NF training on Fmθ, and to examine training-related changes in Fmθ and behavioral performance measures during the Go-NoGo shooting task (% errors of commission, accuracy, RT). Models were constructed in four steps and participants were used as the level 2 grouping variable. Several models were run for each dependent measure: intercepts only models, inclusion of level 1 predictors, inclusion of the level 2 predictors of session number and interactions with session number, and the full model including the level 2 predictors group and group interactions, see equations 1 and 2. Data were analyzed using R (version 4.1.0) in conjunction with several packages: lme4, jtools, ggplot2, and interactions.

EQ1. Equations for full NF models

Level 1: $\hat{y}_{ti} = \pi_{0i} + \pi_{1i}{}^*(\text{Block}) + \pi_{2i}{}^*(\text{Modulation}) + e_{ti}$

Level 2: $\pi_{0i} = \beta_{00} + \beta_{01}{}^*(\text{Session}_i) + \beta_{02}{}^*(\text{Group}) + \beta_{03}(\text{Session}_i{}^*\text{Group}) + r_{0i}$

$\quad\quad \pi_{1i} = \beta_{10} + \beta_{11}{}^*(\text{Session}_i) + \beta_{12}{}^*(\text{Group}) + \beta_{13}(\text{Session}_i{}^*\text{Group}) + r_{1i}$

$\quad\quad \pi_{2i} = \beta_{20} + r_{2i}$

EQ2. Equations for full SH task models

Level 1: $\hat{y}_{ti} = \pi_{0i} + \pi_{1i}{}^*(\text{Condition}) + e_{ti}$

Level 2: $\pi_{0i} = \beta_{00} + \beta_{01}{}^*(\text{Session}) + \beta_{02}{}^*(\text{Group}) + \beta_{03}(\text{Session}_i{}^*\text{Group}) + r_{0i}$

$\quad\quad \pi_{2i} = \beta_{20} + r_{2i}$

Before conducting the modeling, several assumptions related to performing an MLM analysis were examined and outliers were removed. MLM analyses are better able to handle assumptions of independence compared to traditional general linear models but are still subject to many of the same assumptions including normality, linearity, multicollinearity, homogeneity of variances, issues related to residuals, and outliers [52]. Histograms were visually inspected to identify and remove any errors in data entry measurement error. Next, participant scores for each of the outcome measures were considered outliers if they were outside of 2 standard deviations (SD) from the mean. Skewness and kurtosis were then assessed and all were within ±3. Homoscedasticity of the residuals using residual plots and normality of the residuals using QQ plots were examined after the intercepts only models and the full models. Multicollinearity was assessed for the full models and reported as variance inflation factor (VIF) scores.

Fixed effects are reported as unstandardized beta weights with standard errors. Additionally, intraclass correlation coefficients (ICC) were obtained from each model, large ICCs and design effects of greater than 2 provide evidence for multilevel structure to the data. The estimation method used was restricted maximum likelihood (REML). REML tends to lead to more accurate results than full maximum likelihood (FML), when the sample size of the level two identifier is small [53]. Small sample sizes have been associated with inflated type 1 error rates, but a simulation study using MLM found that including at least 30 samples at the second

level is too little bias [52]. A sample size of 30 at level 2 provided reasonable estimates of standard error for level 1 fixed effects but underestimated standard deviations for level 2 predictors [52]. Model fit statistics of Akaike Information Criterion (AIC) and Bayesian Information Criterion (BIC) are provided. Interactions were explored using simple slopes analysis.

**Responder-only analyses.** Responder-only analyses were also conducted, which are reported in Supporting information. For the analyses, participants were considered responders if either session or moderation could predict their Fmθ power during NF training using univariate regression analyses. Using these criteria 18 participants were considered responders and 12 were non-responders.

## Results

### Neurofeedback training

An intercepts only model was conducted, Fmθ ($M$ = -0.18, $SD$ = 2.83), an intraclass correlation coefficient (ICC) was calculated for the model and revealed that 78% of the variance could be accounted for by differences between participants, the design effect from the current analysis was 23.62, AIC = 2981.51, BIC = 2995.66. Block and Modulation were added as level 1 predictors. Adding the level 1 predictors did not meaningfully improve model performance, AIC = 2976.04, BIC = 3004.29, fixed effects pseudo $R^2$ = 0.00, ICC = 78%. Next, Session was added into the model. The model was a better fit to the data compared to the previous model, AIC = 2822.82, BIC = 2865.20, fixed effects pseudo $R^2$ = 0.01, ICC also increased to 86%. Finally, a full model was conducted. The Group x Block, Group x Session, and Group x Session x Block interactions were significant predictors of Fmθ (see Table 1). This model was not a better fit to the data compared to the previous model, AIC = 2831.94, BIC = 2893.15, fixed effects pseudo $R^2$ = 0.04, ICC = 82%. Fig 4 shows Fmθ changes within and across Sessions for each Group. Fig 5 shows FCz full spectrum and topographic maps of theta and alpha power for the first and last training Sessions for each Group, with separate plots for up- and down-modulation in the ALT group. Simple slopes analyses were performed on the statistically significant interactions. It was found that Fmθ increased over Blocks (b = 0.06, p = .04) and Sessions (b = 0.32, p = .04) for the ALT group. This effect was not observed for the INC group for

**Table 1. Full MLM of Fmθ during NF training.**

|  | Fmθ |  |
| --- | --- | --- |
|  | **β** | **_p_** |
| (Intercept) | -0.52 (0.9) | .57 |
| Modulation | 0.10 (0.11) | .33 |
| Block | -0.04 (0.09) | .66 |
| Session | -0.05 (0.17) | .76 |
| Group | -1.08 (1.15) | .36 |
| Block*Session | 0.03 (0.03) | .35 |
| Group*Block | 0.23 (0.12) | **.05** |
| Group*Session | 0.51 (0.22) | **.02** |
| Group*Session* Block | -0.07 (0.04) | **.05** |
| Variance for session slope | 0.45 |  |
| ICC | .85 |  |
| Fixed $R^2$ | .02 |  |
| AIC | 2854.78 |  |
| BIC | 2916.00 |  |

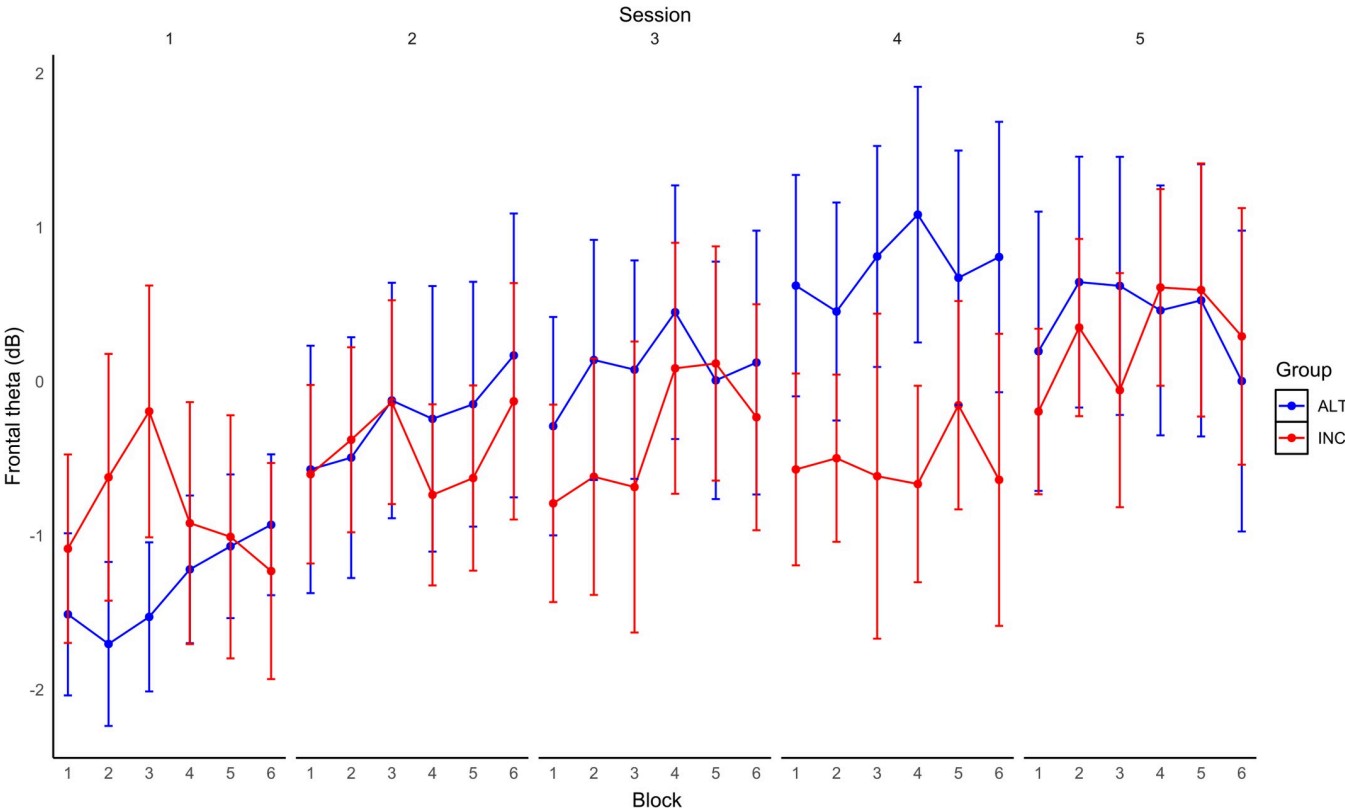

**Fig 4. Fmθ changes within and across sessions for each group during NF.** Changes in Fmθ over Blocks and Sessions of NF training for each Group (error bars are SE).

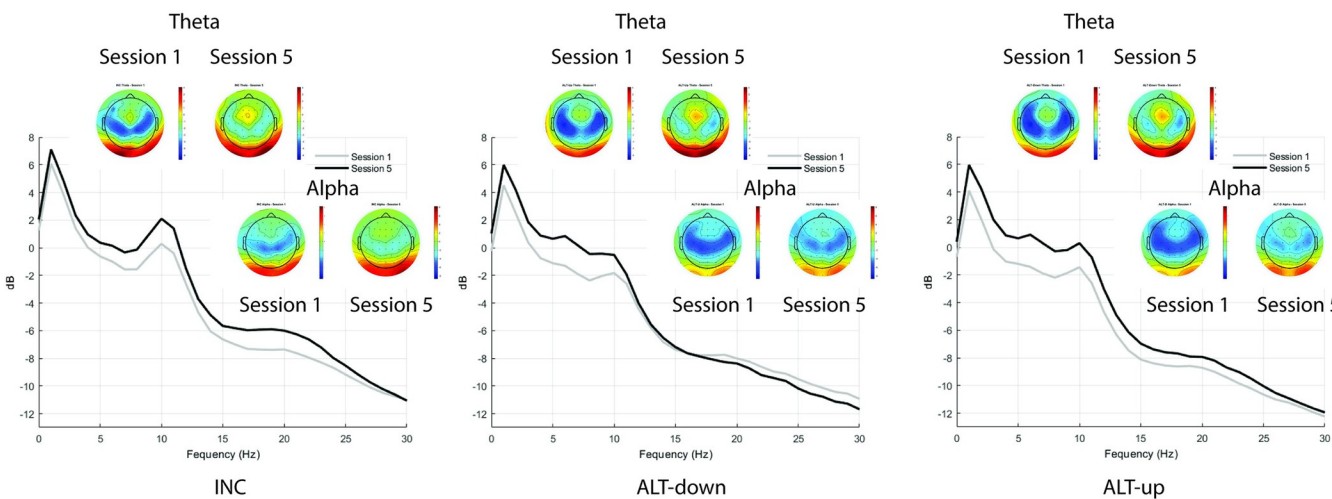

**Fig 5. FCz full spectrum and topographic maps of theta and alpha power for the first and last training sessions for each group during NF.** Topographic maps of mean theta (4–7 Hz; top row) and alpha (8–13 Hz; bottom row) averaged over Blocks for Session 1 (left map) and Session 5 (right map) of NF training and corresponding mean power spectra at FCz from 0–30 Hz for INC group (left spectra), ALT group, down-modulation (middle spectra), and ALT group, up-modulation blocks (right spectra).

Blocks (p = .35) or Sessions (p = .77). The three-way interaction revealed that Fmθ increased for the ALT group over Blocks during Sessions 1 (p = .01), 2 (p < .01), and 3 (p = .04) but not for Sessions 4 or 5 (ps > .05). Fmθ did not change across Blocks for any Session for the INC group (ps > .05). Taken together, only the ALT group evidenced a significant training-related increase in Fmθ (across Blocks), but this effect was limited to the first three sessions of the experiment.

## Go-NoGo shooting task

**Frontal theta.** Fmθ from shooting task trials ($M = 0.85$, $SD = 2.96$) had an ICC = 80% and design effect of 24.20 for the intercepts only model, AIC = 2407.46, BIC = 2420.99. Adding Condition did not meaningfully improve model performance, AIC = 2404.34, BIC = 2422.38, fixed effects pseudo $R^2 = 0.00$, ICC = 81%. Session was then added into the model, improving model fit compared to the previous model, AIC = 2245.87, BIC = 2281.95, fixed effects pseudo $R^2 = 0.00$, ICC also increased to 84%. The full model revealed that none of the fixed effects were statistically significant. Further, this model did not improve model fit, AIC = 2258.07, BIC = 2312.19, the model had a fixed effects pseudo $R^2 = 0.01$, ICC remained at 84% (see Table 2). Fig 6 shows Fmθ changes across Sessions in Low and High time-stress conditions for each Group. Fig 7 shows FCz full spectrum and topographic maps of theta and alpha power for the first and last testing Sessions for each Group. In summary, the ALT and INC groups did not differ with respect to average Fmθ levels or in the degree of Fmθ changes across Sessions or Conditions.

## Behavioral performance

**% Errors of commission.** Examining commission errors ($M = 26.12$, $SD = 16.85$) with an intercepts only model produced an ICC of 29% and design effect of 9.41, AIC = 2710.75, BIC = 2722.11. The level 1 model was a better fit to the data, AIC = 2621.52, BIC = 2636.66. The function had a pseudo $R^2$ of 0.16 for the fixed effects and ICC was 39%. The level 2 model was also a better fit according to AIC but not BIC, AIC = 2618.42, BIC = 2648.72, fixed effects pseudo $R^2 = 0.17$, ICC = 56%. Finally, a full model was conducted. Again this model was a better fit according to AIC but not BIC compared to the previous model, AIC = 2607.26,

**Table 2. Full MLMs of Fmθ during Go-NoGo shooting task.**

|  | Frontal theta | |
|---|---|---|
|  | β | p |
| (Intercept) | 0.63 (0.76) | .41 |
| Condition | 0.14 (0.32) | .65 |
| Session | -0.03 (0.14) | .82 |
| Group | 0.35 (0.98) | .72 |
| Condition*Session | 0.03 (0.08) | .67 |
| Group*Condition | 0.12 (0.40) | .77 |
| Group* Session | 0.00 (0.17) | .99 |
| Group*Condition*Session | -0.03 (0.10) | .77 |
| Variance for session slope | 0.43 | |
| ICC | .84 | |
| Fixed $R^2$ | .01 | |
| AIC | 2258.07 | |
| BIC | 2312.19 | |

Condition

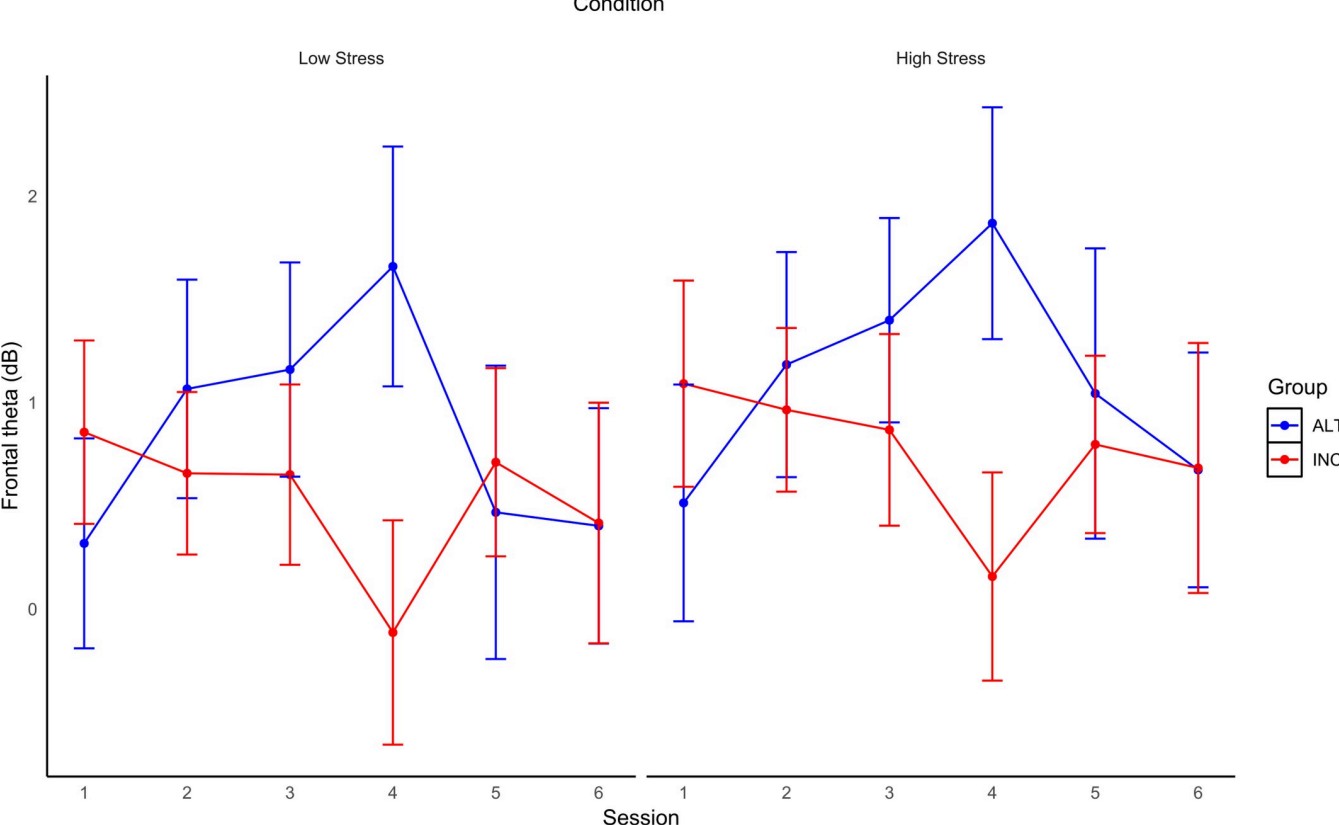

**Fig 6. Fmθ changes across sessions in low and high time-stress conditions for each group during SH.** Changes in Fmθ over Sessions during Go-NoGo shooting task in Low (left) and High (right) time-stress conditions for each Group (error bars are SE).

BIC = 2652.70. The function had a pseudo $R^2$ of 0.19 for the fixed effects and ICC was 57%. Session and Condition were statistically significant predictors of the number of commission errors performed (see Table 3 and Fig 8). These results indicate that commission errors were higher under High relative to Low time-stress and that commission errors (across both Low and High time-stress conditions) declined across Sessions.

**Accuracy.** The intercepts only model for accuracy ($M = 66.12$, $SD = 17.22$) had an ICC = 10% and the design effect was 3.90, AIC = 2893.97, BIC = 2905.46. Next, Condition was added as a level 1 predictor and the model improved fit indices, AIC = 2531.28, BIC = 2546.60, fixed effects pseudo $R^2$ = 0.56, ICC = 39%. The level 2 model was also a better fit to the data compared to the level 1 model, AIC = 2462.63, BIC = 2493.26, fixed effects pseudo $R^2$ = 0.60, ICC = 57%. The full model was a better fit according to AIC but not BIC, AIC = 2452.73, BIC = 2498.67, fixed effects pseudo $R^2$ = 0.60, ICC = 57%. Session, Condition, and the Group x Session interaction were significant predictors of accuracy (see Table 3 and Fig 9). Simple slopes analysis revealed that accuracy increased over Sessions for both the INC group ($b = 3.16$, $p < .01$) and for the ALT group ($b = 1.64$, $p < .01$). These results indicate that accuracy increased over Sessions across all participants and that the magnitude of this increase did not differ between the NF conditions.

**Reaction times.** Reaction time ($M = 485.05$, $SD = 76.36$) produced an intercepts only model with an ICC of 49%, the design effect was 15.21, AIC = 3723.03, BIC = 3734.48. Adding the level 1 predictors improved model performance, AIC = 3424.31, BIC = 3439.58, fixed

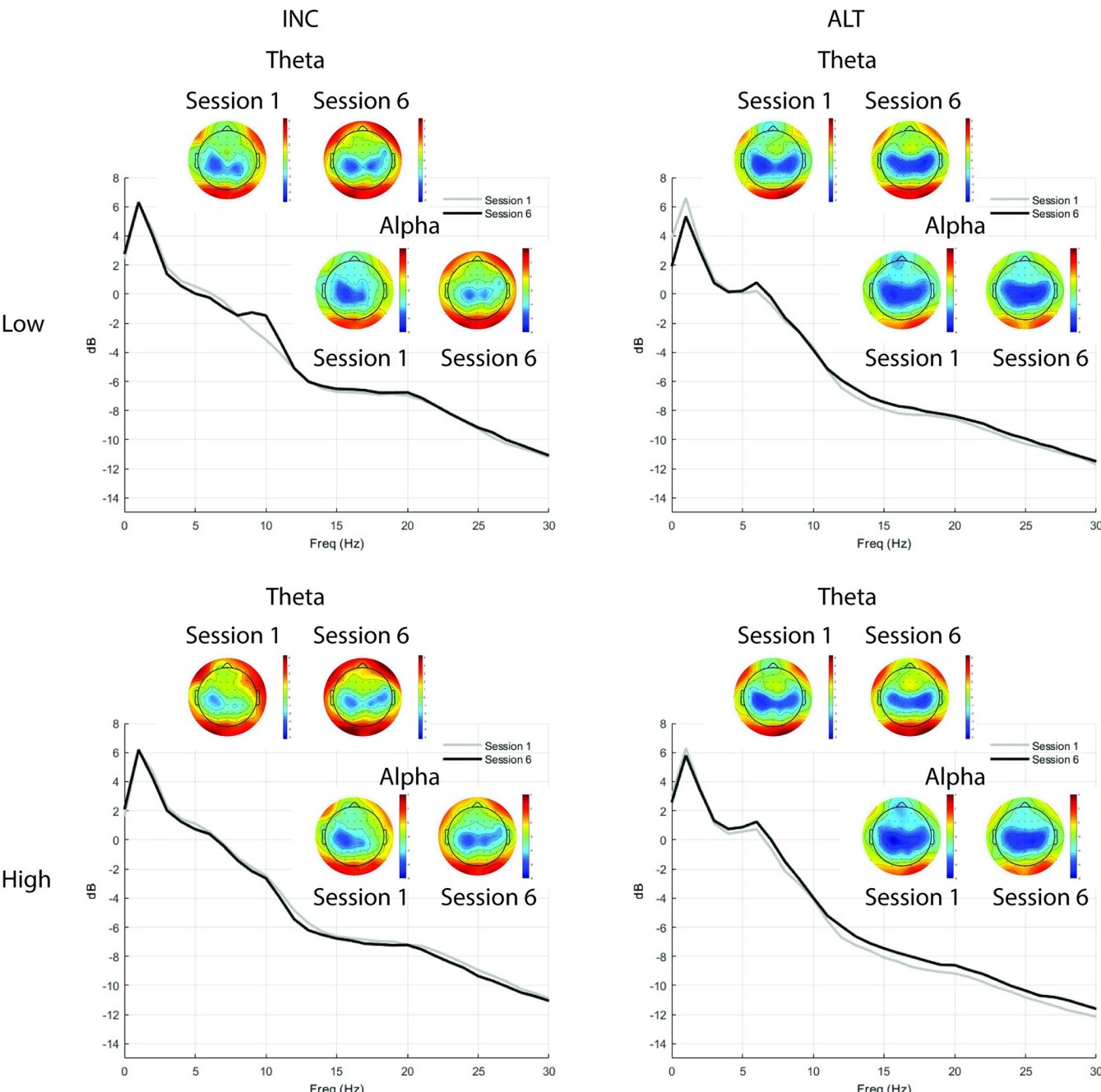

**Fig 7. FCz full spectrum and topographic maps of theta and alpha power for the first and last testing sessions for each group during SH.** Topographic maps of mean theta (4–7 Hz; top row) and alpha (8–13 Hz; bottom row) power averaged over Blocks for Session 1 (left map) and Session 6 (right map) during SH task and corresponding mean power spectra at FCz from 0–30 Hz for INC group (column 1) and ALT group (column 2) in Low (row 1) and High (row 2) time stress conditions.

effects pseudo $R^2$ = 0.28, ICC = 74%. The level 2 model accounted for a greater portion of variability than the level 1 model, AIC = 3406.99, BIC = 3437.53, fixed effects pseudo $R^2$ = 0.28, ICC = 79%. A full model was conducted, the Condition as well as the interactions of Session x Condition, Group x Condition, and Group x Session x Condition were statistically significant. AIC indicated better model fit but BIC did not, AIC = 3385.99, BIC = 3431.80, fixed effects

**Table 3. Full MLMs for behavioral measures from Go-NoGo shooting task.**

| | % Commission errors | *p* | Accuracy | *p* | Reaction time | *p* |
|---|---|---|---|---|---|---|
| | β | | β | | β | |
| (Intercept) | 26.64 (5.13) | < .01 | 66.37 (3.32) | < .01 | 559.13 (20.47) | < .01 |
| Condition | 11.84 (4.89) | **.02** | -24.47 (3.19) | < .01 | -124.21 (12.97) | < .01 |
| Session | -2.32 (1.01) | **.02** | 3.68 (0.73) | < .01 | -5.74 (3.04) | .06 |
| Group | -5.95 (6.61) | .37 | 5.57 (4.29) | .20 | -26.96 (26.38) | .31 |
| Condition*Session | 0.38 (1.24) | .76 | -1.07 (0.82) | .19 | 10.65 (3.31) | < .01 |
| Group*Condition | 2.18 (6.30) | .73 | -0.36 (4.05) | .93 | 37.21 (16.59) | **.03** |
| Group*Session | 2.32 (1.29) | .08 | -1.99 (0.93) | **.04** | 2.32 (3.90) | .55 |
| Group*Session*Condition | -0.17 (1.6) | .92 | 0.98 (1.03) | .34 | -9.35 (4.22) | **.03** |
| Variance for session slope | 1.79 | | 1.61 | | 6.62 | |
| ICC | .57 | | .57 | | .80 | |
| Fixed $R^2$ | .19 | | .60 | | .29 | |
| AIC | 2607.26 | | 2452.73 | | 3385.99 | |
| BIC | 2652.70 | | 2498.67 | | 3431.80 | |

pseudo $R^2$ = 0.29, ICC increased to 80% (see Table 3 and Fig 10). Finally, simple slopes analyses indicated that reaction time decreased over Sessions in the Low (b = -4.32, p = .03) but not the High time-stress condition (p = .74). Reaction times also decreased over Sessions for both

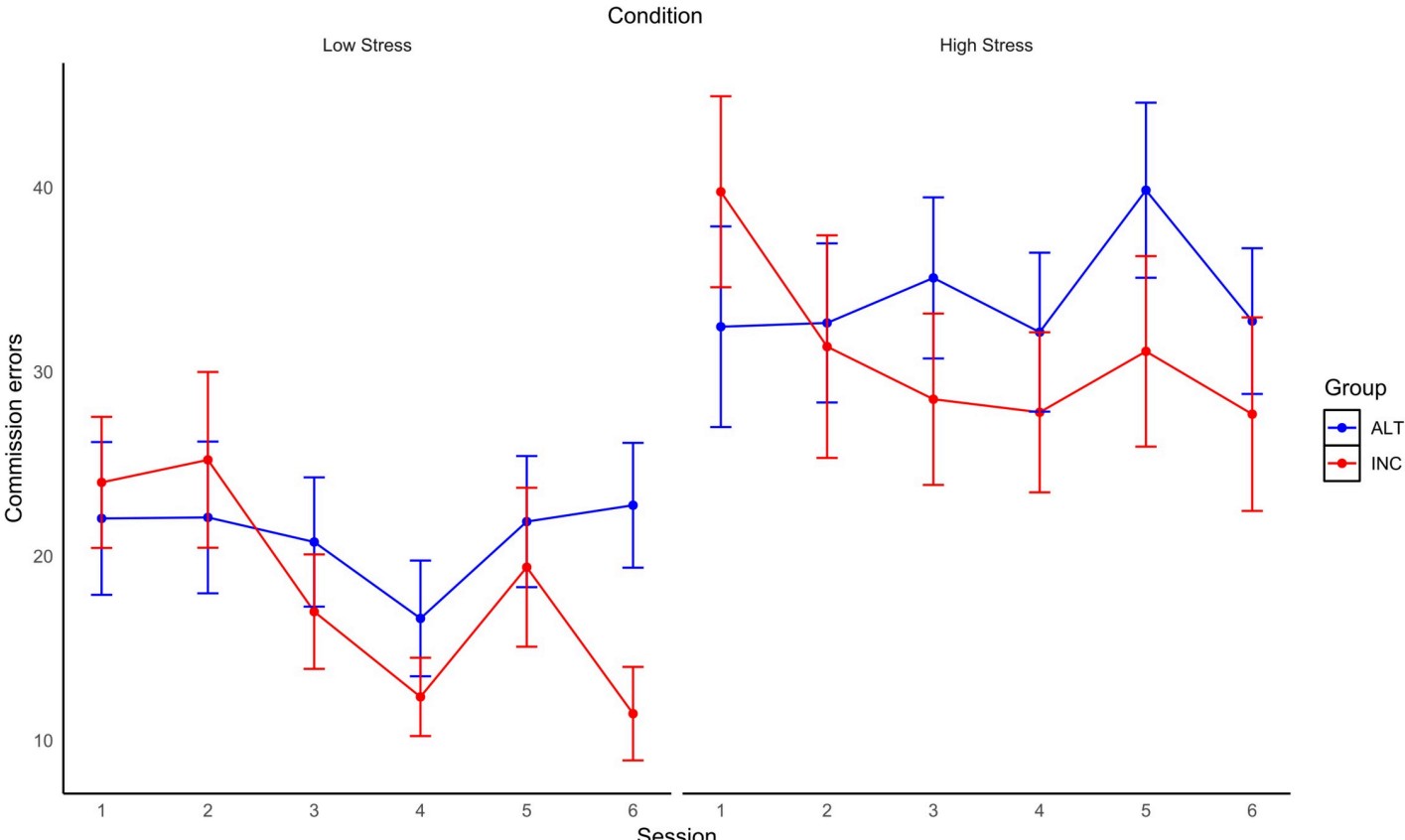

**Fig 8. Changes in errors of commission across sessions in low and high time-stress conditions for each group.** Percentage errors of commission over Sessions in the Low (left) and High (right) time-stress conditions for each Group (error bars are SE).

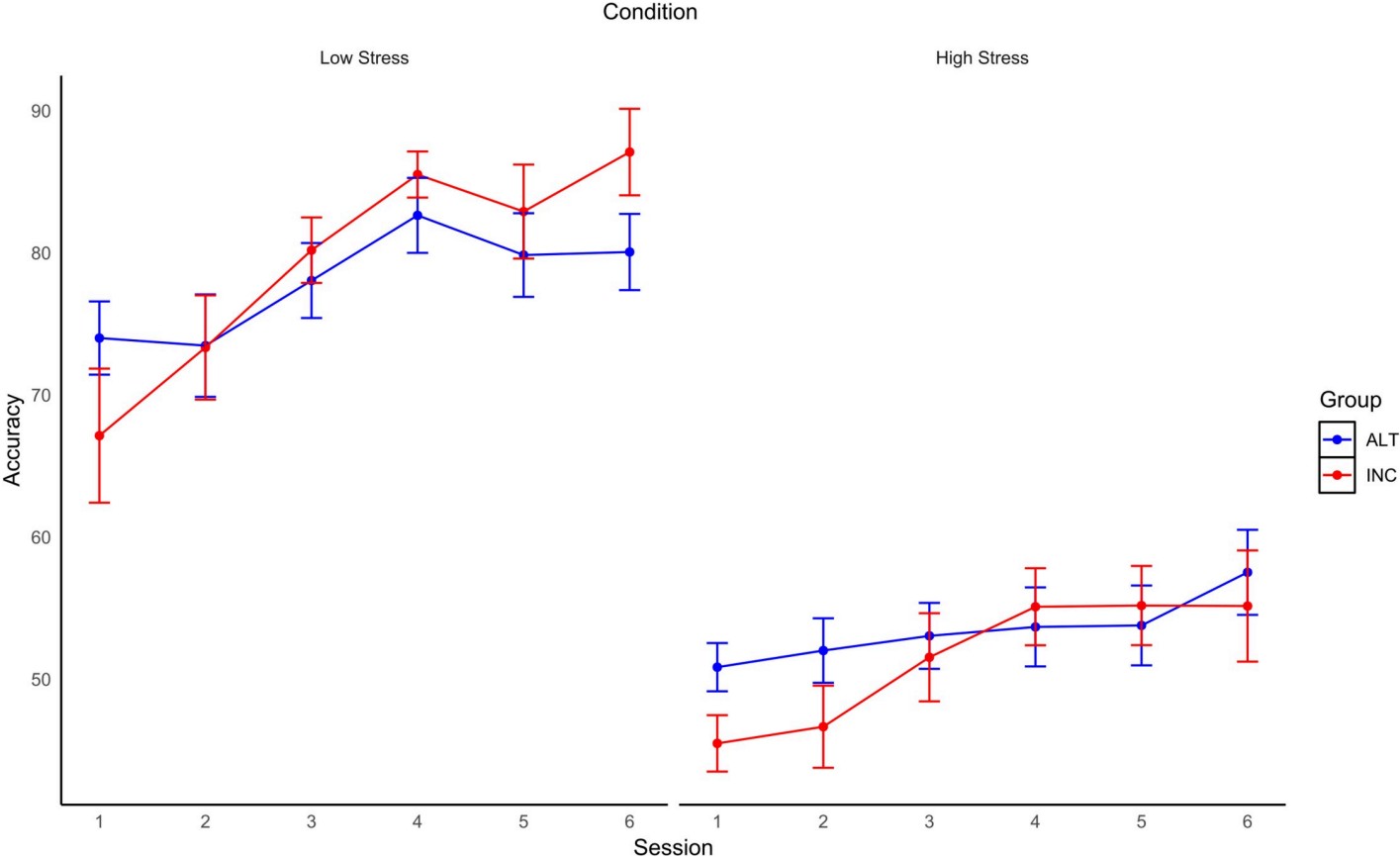

**Fig 9. Changes in shooting accuracy across sessions in low and high time-stress conditions for each group.** Percentage enemy targets hit over Sessions in the Low (left) and High (right) time-stress conditions for each Group (error bars are SE).

the INC group (b = -86.47, p < .01) and the ALT group (b = -82.39, p < .01). Reaction times decreased among INC group participants in the Low stress condition over Sessions (b = -5.47, p = .06) but not in the High stress condition, whereas they did not decrease over Sessions for the ALT group (all p > .05). Taken together, these findings indicate that the INC group decreased RTs in the Low, but not High, time-stress condition, whereas RTs in the ALT group did not change over Sessions in either Condition.

## Discussion

We investigated neurobehavioral effects of a novel NF training protocol that targeted both up- and down-modulation of Fmθ. This novel alternating up/down modulation condition (ALT) was compared to a more common NF training condition targeting solely up-modulation of Fmθ (INC). Accurate contingent feedback was provided in both groups; only the top-down goals of the training differed between groups, such that ALT group required switching of top-down training goals whereas the INC group required maintenance of a consistent top-down training goal. Differences between groups were investigated both in terms of (1) NF training-related changes in Fmθ and (2) changes in Fmθ and performance outcomes during a Go-NoGo shooting task. Overall, the present findings indicate greater Fmθ increases across training sessions in the ALT versus INC group. Accuracy and reaction time metrics improved at a steeper rate in the INC relative to the ALT group. Taken together, these results are consistent

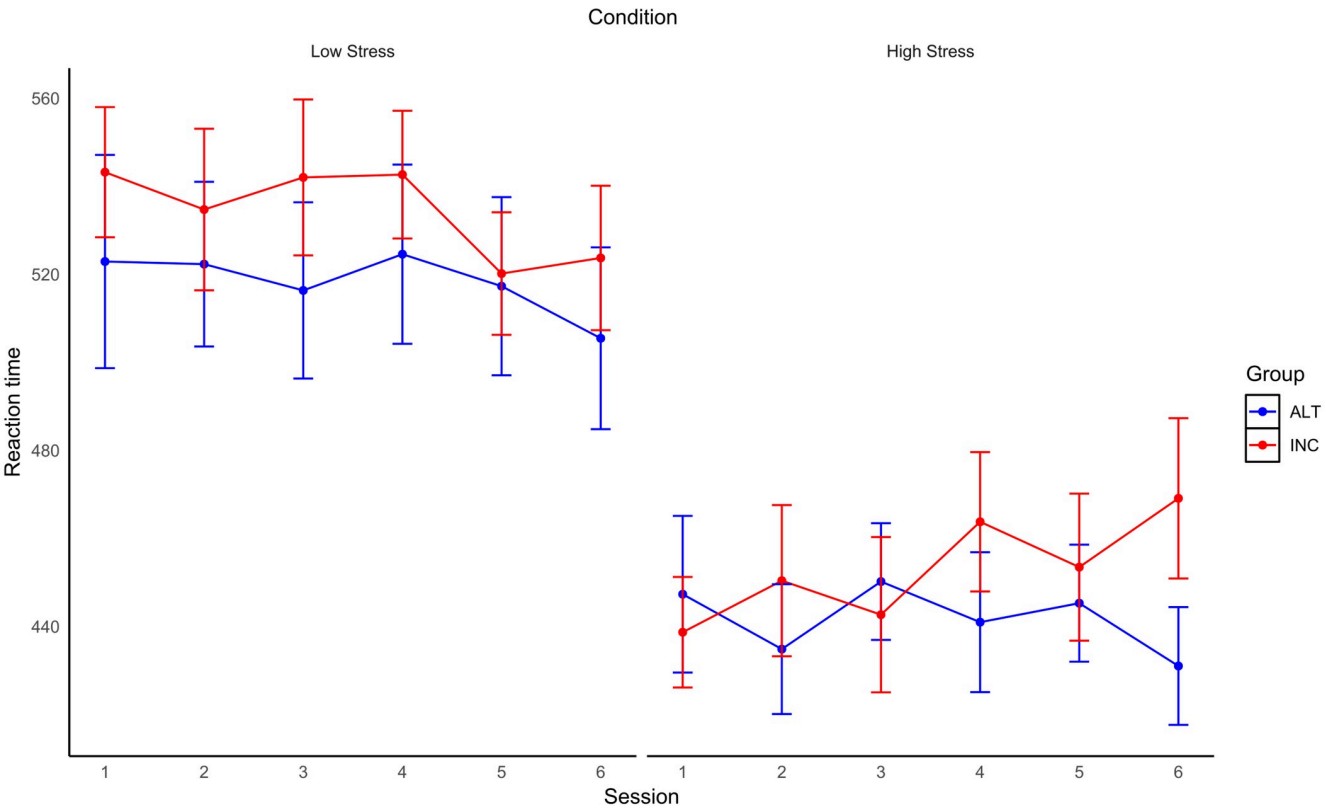

**Fig 10. Changes in RTs across sessions in low and high time-stress conditions for each group.** RTs to enemy targets over Sessions in the Low (left) and High (right) time-stress conditions for each Group (error bars are SE).

with the notion that alternating up- and down-modulation of Fmθ incurred switching costs that led to overall increases in theta (across sessions). It should be noted that these inferences are merely suggestive in light of insufficient number of sessions to explore alternative accounts of the findings (e.g., learning). Future studies should corroborate these findings using more practice sessions. Nevertheless, this study is an important step in unraveling the breadth of complex neurocognitive mechanisms at play in NF protocols.

One question about using sham control groups, the gold standard in NF training research [4], is that it's difficult to unambiguously attribute observed differences in the training-related changes in brain activity or performance outcomes in the real versus sham feedback groups to accurate contingent feedback *per se*. In sham control studies, it is assumed that reinforcement learning afforded by accurate contingent feedback in the real NF training group enables learning of self-regulation, while the lack of accurate contingent feedback in sham controls precludes it. Further, it assumes that self-regulation processes afforded by accurate contingent feedback and non-specific executive control processes such as attention and effort are mutually exclusive. Finally, it assumes that because Fmθ increases during performance of executive tasks, increasing it through NF training will generalize to increased Fmθ and performance of executive tasks. However, it's still not clear whether effects observed in previous research are due to increased activation associated with self-regulation based on accurate contingent feedback, or some other mechanism(s). For example, sham participants may eventually discover that the feedback does not seem to correspond to their strategies or efforts to try to modulate it. Brandmeyer & Delorme [23] reported that 8 of 12 participants in the sham control group

perceived a lack of correspondence between the successful implementation of their strategies and the feedback of the visual stimulus. In contrast, all 12 participants in the real feedback group reported perceived correspondence between strategies and feedback. So, what is the effect of decoupling accurate contingent feedback from top-down goals in sham control studies? False feedback may lead to sham participants becoming bored and disengaged from the task, or it may encourage them to inconsistently explore ineffective alternative strategies while never persisting in any one strategy or having any possibility of learning the task without contingent feedback. On the other hand, another potential confound of such sham control conditions is that attentional engagement and cognitive control strategies of sham participants (whether suspect or not as to the veracity of the feedback) may differ little if at all from those who receive real feedback (this is the general assumption of using sham control). Thus, it's difficult to know whether differences observed between real and sham control groups in previous research (in either EEG or behavior outcomes) are due to differential levels of task engagement or cognitive control strategies, or whether the lack of accurate contingent feedback precludes learning of self-regulation and underlies differential training effects regardless of attentional engagement or strategy.

In the present study, the difference in training between ALT and INC groups was the top-down goals of increasing, or alternating increasing and decreasing Fmθ. Considering that real contingent feedback was provided in both groups, the coupling of top-down and bottom-up error-related processing of visual stimuli should be comparable between groups (i.e., both groups should be able to self-regulate Fmθ based on accurate contingent feedback). In previous research where up-modulation of Fmθ groups were compared to sham control groups receiving false or non-contingent feedback, the opposite could be said. That is, top-down goals were the same between groups (up-modulation), but the coupling with bottom-up error-related processing demands would be artificially altered in sham control group subjects because the lack of correspondence between top-down goals and bottom-up feedback. On the other hand, because experimental and sham control groups receive the same visual stimuli (when pairs of subjects are matched), they experience identical bottom-up visual feedback (except where artifacts are produced). The difference would be the natural coupling of top-down and bottom-up attention signals in experimental group, but decoupling in the sham group. According to Corbetta and Shulman [54], coordination between dorsal and ventral frontal-parietal attention networks is crucial for maintaining current (top-down) goals, while monitoring (bottom-up) stimulus-driven attention processes from moment-to-moment, respectively. Thus, false or non-contingent feedback may disrupt the dorsal and ventral frontal-parietal attention network dynamics and thus the capability to learn the task.

## Neurofeedback training

In an attempt to disentangle confounding effects of training-related increases in Fmθ observed in previous research in real NF versus sham control groups, i.e., self-regulation based on contingent feedback, or other mechanisms, we tested two competing hypotheses while holding accurate contingent feedback constant between groups. First, if Fmθ increases are greater in the INC group, it would suggest that the effects of increased activation are due to reinforcement learning of self-regulation afforded by accurate contingent feedback, in support of previous research. Second, if Fmθ increases are greater in the ALT group, it would suggest that the effects of increased activation are not due to due to self-regulation, as ALT participants should exhibit lower Fmθ, especially during down-modulation blocks. Rather, greater increased activation in the ALT group may be associated with greater task demands imposed by having to switch between top-down goal states of up- and down-modulation (i.e., switch-costs). Support

for the latter hypothesis would call into question previous research suggesting that Fmθ increases observed in NF training protocols designed to up-modulate Fmθ are unambiguously due to reinforcement learning of self-regulation. Based on the significant Group x Session x Block interaction observed in the present study, we found support for the hypothesis that the effects of increased activation in the ALT versus INC group within and across NF training sessions are not due to reinforcement learning of self-regulation afforded by accurate contingent feedback. According to Womelsdorf, Vinck, Leung, and Everling [16], phasic theta synchronization is associated with reactivation of contextual task rule representations (stimulus–response mappings) and ongoing assessment of sensory evidence. Thus, one possible explanation for why the ALT group exhibited increased Fmθ in the present study is that a task (goal)-switch cost was imposed on the ALT, but not the INC group. Task-switching imposes greater executive control demands than maintaining a consistent task set [55], requiring reactivation of task set rules, which may account for the higher Fmθ observed in the ALT but not INC group. However, these interpretations must be tempered based on one difference observed in the responder-only analysis (Supporting information). That is, a main effect of Group was observed such that participants in the INC group exhibited higher Fmθ than the ALT group. These findings must also be interpreted with caution because of the loss of statistical power when analyses are conducted on a subset of the data (n = 18). Also, considering the relatively small number of training sessions, it is unlikely that sufficient training was provided in the present study.

We're not aware of any studies on Fmθ NF which have directly compared up- vs. down-modulation training protocols to each other in the same study in young healthy participants. Our results suggest that Fmθ is involved in the attempted active modulation of brain states when accurate contingent feedback is provided when attempting to alternate increases and decreases within the same session, whether the top-down goal is to increase or decrease it. According to meta-analytic fMRI NF research, Emmert et al. [45] reported that, "independent of the outcome of the neurofeedback, a wide network of areas involved in cognitive control and sensory processing is recruited during attempted self-regulation" (p. 810). In the present study, the contradictory finding that Fmθ increased in the ALT group, regardless of attempted up- or down-modulation, is a novel finding in EEG NF research but is consistent with [45]. The likewise contradictory lack of significant Fmθ increases in the INC group suggests that Fmθ, and by inference network dynamics, also differ depending on the dynamic switching of top-down goals during NF training. Enriquez-Geppert et al. [19, 24] and Brandmeyer & Delorme [23] showed increased Fmθ in the training group over the first 5 sessions, and then a leveling off over sessions 6–8, whereas Wang and Hsieh [21] showed linear increases over 12 sessions. It's possible that the volume of training in our study (18 min / session) was not sufficient to induce significant changes relative to previous studies (30 min / session), or perhaps the different structures of the training sessions (six 30 s trials per block versus 5-min continuous blocks) may account for differences. However, such differences in training did produce significant effects for the ALT group, suggesting that the volume and structure of the NF training protocol was sufficient to induce training-related effects when the top-down goals changed randomly between blocks within and across sessions, although these changes appear to be unrelated to reinforcement learning of self-regulation from NF training. It is noteworthy also to mention here, that an unexpectedly high percentage of 1-s epochs during 30-s modulation trials (~27%) was rejected due to artifact. No differences were revealed between groups (F(1, 857) = 2.33, p = .13) or changes over sessions (F(4, 849) = 0.94, p = .44). This observation suggests that the participants in both groups, whether intentionally or inadvertently, found it challenging to control the signal with just their mental strategies or focus of attention.

## Go-NoGo shooting task performance

Regarding training-related performance outcomes, we hypothesized that the ALT group would improve at a greater rate on the Go-NoGo shooting task over sessions, as the dynamics-to-function mapping of activation and inhibition attempted in training would be more specific to the demands of the Go-NoGo task to which the desired training outcome should map onto [1]. In particular, this effect should be observed for errors of commission, where the top-down goals of activating or inhibiting a response are challenged. This hypothesis assumes a feedback-related self-regulation effect in which the participants are able to better regulate the activation or inhibition of Fmθ as a function of NF training. However, this hypothesis was rejected as no differences were observed between groups for changes in errors of commission; both groups showed comparable decreases in the percentage of errors of commission in both time-stress conditions. For shooting accuracy at enemy targets (Go responses), the Group x Session interaction revealed that the INC group exhibited greater improvements over sessions. However, in the responder-only analysis, this effect was non-significant. For reaction time to enemy targets (Go responses), the Group x Session x Condition interaction revealed differences in reaction times between groups over sessions and conditions. Specifically, the INC group showed marginal (p = .06) reductions in reaction times in the Low time-stress condition, an effect not observed in the ALT group. Thus, although Fmθ did not increase significantly in the INC group over NF training sessions, whereas it did in the ALT group, performance improvements increased at a significantly lower rate in the ALT versus INC group with respect to accuracy and reaction time, but did not differ with respect to inhibitory control. However, in the responders-only analysis, interaction effects were non-significant. Collectively, these findings contradicted our hypotheses, that the ALT group would exhibit greater improvement on the Go-NoGo task as a function of greater training specificity with respect to altering activation and inhibition. Considering the relatively higher Fmθ activation observed over NF training sessions in the ALT versus INC group, together with the equivocal or lesser performance gains depending on which measure of performance we examine, our results suggest that performance differences between groups are unlikely driven by either increased activation or learned regulation of Fmθ. So how can these seemingly paradoxical findings be reconciled?

Previous research has shown that, although Fmθ increases in NF relative to sham control groups, performance on inhibitory control or conflict detection tasks (Stroop, stop-signal, local-global task) did not improve pre-post training for either NF training or sham control groups [19, 23]. On the other hand, several studies have shown that Fmθ NF training groups performed better on working memory tasks vs. sham groups post-training [19, 21, 23]. Considering that working memory functions operate over longer time scales (encoding and retrieval operations) than transient Go-NoGo or Stroop trials, it may be that training designed to tonically increase Fmθ over relatively long periods (e.g., 5 min trial blocks) is more conducive or specific to memory functions than inhibitory control or conflict monitoring functions, which operate over shorter timescales (ms). We attempted to take this into consideration in the present study by implementing shorter discrete trials of NF modulation (30 s), and by including the ALT group which was instructed to alternate activation and inhibition of Fmθ activity randomly over blocks. However, on the timescales of phasic EEG dynamics and ms timescale task dynamics, these design changes in NF training may still have been operating on longer timescales or inducing different dynamic attention states (i.e., perhaps inconsistent dynamics-to-function mapping from NF training to SH task [1]. In the future, researchers should conceive of designs featuring NF training protocols that could be implemented on a more phasic timescale which is more consistent with the brain dynamics underlying the tasks

targeted for behavioral improvement. For example, it would be interesting to test whether feedback designed to maintain instability or criticality of ongoing EEG [56, 57], rather than sustained synchronization of oscillatory EEG frequencies would better translate to improved executive task outcomes.

In the present study, we implemented NF training within the same session before Go-NoGo task testing, except for the first (pre-) and last (post-) training sessions, whereas previous research conducted NF training in separate sessions between pre- and post-task performance evaluation sessions [19, 21, 23]. Therefore, our results may be explained by the possibility that alternating between increasing and decreasing Fmθ in the ALT group is more cognitively demanding, which in turn may have led to higher levels of fatigue and lower performance gains during the Go-NoGo task. As such, instead of increasing performance gains (as we hypothesized), ALT may thus have weaker performance gains relative to INC due to cognitive fatigue and/or limited cognitive capacity after the up/down task switching protocol. In this study, we tested task performance pre-training, as well as within each training session, and post-training to better characterize the nature and extent of performance changes associated with NF training. However, this difference in study design and the possibility of fatigue-related effects makes it difficult to directly compare results with previous studies.

It is interesting to note in Figs 5 and 7, that FCz spectra shows peaks at 6 Hz in the ALT group during both up- and down-modulation in NF training and also during the Go-NoGo task, but is absent in the INC group in both NF and Go-NoGo. Also, in Figs 4 and 6, it can be seen that FCz theta increases linearly over sessions 1–4 during both NF training and Go-NoGo shooting task performance in the ALT group, which lends support to a possible fatigue-related effect attributable to NF-related Fmθ increases. Further, the ALT group shows a less prominent alpha peak and narrower width than the INC group during NF training, whereas both groups show relatively complete suppression of alpha during the Go-NoGo task (Figs 5 and 7). Alpha suppression is also consistent with greater task demands, which supports the interpretation that the NF training demands associated with task (goal)-switching in the ALT group imposed a relatively higher executive control demand. These differences in features in theta and alpha peaks suggest differences in oscillatory dynamics within theta and alpha frequencies in the frontal region, as well as coupling across regions and frequencies throughout the cortex during NF training, perhaps with transference to the Go-NoGo task with respect to the existence (ALT group) or non-existence (INC group) of frontal theta peaks. Cross-frequency coupling and network analyses would shed further light on these outstanding questions.

Finally, few studies have examined brain network adaptations associated with EEG NF training but novel methods and approaches are rapidly advancing to investigate the nature of dynamic functional interactions among different brain regions and adaptations associated with NF training, particularly in the fMRI literature [44, 45, 58–62]. Enriquez-Geppert, Huster, and Hermann [63] proposed three main patterns of change in neural activity that can be expected in response to NF training: (1) decreased activation of nodes in the network or reduced spatial extent of network, (2) increased activation of nodes in the network or spatial expansion of the network, and (3) functional restructuring in the form of either redistribution (i.e., changes in relative contribution of specific areas while overall patterns are unchanged) or reorganization (i.e., changes in overall network patterns). Haller et al. [44] reported that NF is "mediated by widespread changes in functional connectivity" (p. 243) and Gruzelier [25] reported that "a frontal locus is the strongest locus showing changes following training, though this is not an exclusive locus. Even posterior training predominantly leads to stronger frontal than posterior effects. . . The frontal locus may well represent top-down thalamo-frontal EEG regulatory influences (e.g., [64]), hypothesised speculatively here to follow the mastery of the learned self-regulation" (p. 167). Further, as stated by Papo [1], "the consequences of targeting

a spatially local region can be, and indeed generally are, complex and spatially non-local and multiscale in the anatomical space" (p. 4). Future research should investigate network-based analyses, particularly graph theoretic approaches, to more comprehensively examine EEG NF-related network changes within and across sessions.

## Limitations and future directions

A major limitation of this study was that we did not include a third sham control group, and/ or a down-regulation only NF training group. This was mainly because of practical concerns of time (for both experimenters and participants) and cost. Ideally, we would have included a sham control group and an exclusive down-modulation group. Additionally, the duration of training sessions, the structure of NF trials, and the number of sessions may also have been insufficient to elicit robust training effects. However, reducing NF trial durations from continuous 5 min or more in many studies to 30 s trials was to minimize the likelihood of boredom and fatigue and maximize operant conditioning principles of learning [31]. We also intended to better map the temporal dynamics of NF modulation during training to the temporal dynamics of brain dynamics during the Go-NoGo task [1], while still enabling sufficient time for the participants to gain control of the feedback signal. Also, we took into consideration the idea of task-specific training in terms of temporal dynamics of phasic versus tonic effects, which required a tradeoff between enabling enough time for participants to gain control of their Fmθ, while also keeping trials sufficiently short to elicit more phasic vs. tonic dynamics. Similarly, because NF training + SH task testing sessions were relatively long (~2.5–3 hrs), boredom and/or fatigue may have been an issue in this study. However, by conducting performance tests in close proximity to NF training, this approach enabled us to more closely temporally link NF training to task testing across timescales from within sessions to across sessions, and to examine performance changes with finer temporal resolution beyond just pre-post changes. Further, because NF is a highly attention-demanding task that must be learned through extended deliberate practice like many skills, future research should more systematically explore training parameters such as volume, duration, intensity, frequency/distribution, and variation/individualization of trials and sessions (see [30]). Finally, future research should consider behavioral testing after a longer period of delay (e.g., weeks or months [22, 26]) to assess whether consolidation of training effects are observed without possible confounds of fatigue when measured in the same session.

## Summary and conclusions

In the present study, the difference between ALT and INC training groups was task-switching of top-down goals of increasing or alternating increasing and decreasing Fmθ activity. Considering that accurate contingent feedback was provided in both groups, coupling between top-down goals and bottom-up error-related processing of visual feedback stimuli was comparable between groups. Our results are difficult to form any firm conclusions given its limitations. However, we hope that we have introduced a proof-of-concept based on new ideas to better stimulate future research based on the novel concepts offered. In particular, protocols designed to test greater outcome specificity, in terms of more specific dynamics-to-function mapping [1] between training and task outcomes should be explored, as well as testing against various control conditions beyond just sham control [30]. Along these lines, temporal dynamics (tonic vs phasic) and nonstationarity of EEG should also be given further consideration in future research and novel approaches should be applied to investigate alternative feedback signals or features that may be used beyond spectral analyses (e.g., mutual information, entropy, complexity, correlation dimension [65, 66]. In addition, novel measures should be explored, for

example nonlinear analyses to determine whether changes in complexity or mutual information exchange are affected by NF training and may reflect more suitable control parameters [1]. Because NF training adaptations involve whole brain networks, spanning multiple frequency bands, future research should investigate cross-frequency coupling [67] and network analyses [68] to further explore more holistic brain adaptations to NF training with respect to different frequency bands and topographic brain regions. The effects of NF training on coupling among neural, physiological, and cognitive/behavioral response systems should be considered from a more holistic, complex systems perspective (e.g., see [69]). From a complex systems perspective, assumptions about differentiation (i.e., integration/segregation) of multiple concurrent executive functions (self-monitoring, attention, conflict monitoring, action control, decision making, memory, etc). should be more carefully considered and challenged whether sham control or alternative control conditions are employed in NF training research. Finally, more training sessions should be provided, and follow-up studies are needed to better understand longer-term effects of NF training [22].

## Supporting information

**S1 Fig. Responder-only changes in Fmθ over sessions during NF training and Go-NoGo shooting task and changes in errors of commission, accuracy, and RT over sessions during Go-NoGo shooting task.**
(DOCX)

**S1 Table. Full MLM for responder-only of Fmθ during NF training and Go-NoGo shooting task and errors of commission, accuracy, and RT over sessions during Go-NoGo shooting task.**
(DOCX)

**S1 Text. Results of responder-only MLM analyses of Fmθ during NF training and Go-NoGo shooting task and errors of commission, accuracy, and RT over sessions during Go-NoGo shooting task.**
(DOCX)

## Acknowledgments

We gratefully acknowledge Michael Hammond, Zheng Li, and Vikramaditya Battina for engineering system design and testing, and Theo Feng for scheduling, data collection, and study administration. We also thank the peer reviewers for their valuable contribution to this manuscript.

## Author Contributions

**Conceptualization:** Scott E. Kerick, Derek P. Spangler, Justin B. Brooks, Javier O. Garcia, Nilanjan Bannerjee, Ryan Robucci.

**Data curation:** Scott E. Kerick, Nilanjan Bannerjee, Ryan Robucci.

**Formal analysis:** Scott E. Kerick, Justin Asbee, Derek P. Spangler, Thomas D. Parsons.

**Funding acquisition:** Nilanjan Bannerjee, Ryan Robucci.

**Investigation:** Scott E. Kerick, Justin B. Brooks, Javier O. Garcia, Nilanjan Bannerjee, Ryan Robucci.

**Methodology:** Scott E. Kerick, Justin B. Brooks, Javier O. Garcia, Nilanjan Bannerjee, Ryan Robucci.

**Project administration:** Scott E. Kerick, Nilanjan Bannerjee, Ryan Robucci.

**Resources:** Scott E. Kerick, Nilanjan Bannerjee, Ryan Robucci.

**Software:** Nilanjan Bannerjee, Ryan Robucci.

**Supervision:** Scott E. Kerick, Justin B. Brooks, Nilanjan Bannerjee, Ryan Robucci.

**Validation:** Scott E. Kerick.

**Visualization:** Scott E. Kerick, Justin Asbee.

**Writing – original draft:** Scott E. Kerick, Justin Asbee, Derek P. Spangler, Thomas D. Parsons.

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
