## [Decision Letter · Decision Letter 0]

10 Jan 2022

PONE-D-21-31557Neural and behavioral adaptations to frontal theta neurofeedback training

PLOS ONE

Dear Dr. Kerick,

Thank you for submitting your manuscript to PLOS ONE. After careful consideration, we feel that it has merit but does not fully meet PLOS ONE’s publication criteria as it currently stands. Therefore, we invite you to submit a revised version of the manuscript that addresses the points raised during the review process.

1) In lines 54-55 authors postulated that neurofeedback (NF) "is a highly attention-demanding task that must be learned through extended deliberate practice like many skills." This is not in accordance with the fact that in their experiment they only give 5 sessions. This should be discussed. Even in operant conditioning learning that compromises survival, such a small number of sessions does not make learning happen; It is difficult then to believe that it happens when life is not threatened ...

2) lines 82-100: Up-regulation of Fmθ has been used to improve the cognitive performance in healthy young subjects. Up-regulation of Fmθ protocol should not be confused with the protocol to promote REDUCTION in theta power, used by Becerra et al. (2012), in which the subjects were older adults at risk of cognitive decline (evidenced by an excess of theta power for their age, when compared with normative values). Please review the rationale of this last paper.In this study, a lead was selected in each subject to give the NF. Although in 7 of 14 subjects a frontal lead was selected (in 3 from the experimental group and in 4 from the control group), Fz was never selected; i.e., no subject was down-regulated Fmθ. Therefore, the sentence in lines 98-100 ("Taken together, there is preliminary evidence that either up regulating or down-regulating Fmθ can potentially promote gains in executive performance of some tasks) is not true.A crucial point in this regard is that these older adults had excess theta at rest, and must be undergoing some compensatory process to correct this abnormality, since their cognitive performance is normal; while the Fmθ protocol of the other studies mentioned has been applied mainly to young subjects for whom the normality / abnormality of their EEG is unknown. 3) lines 105-108: Considering that accurate contingent feedback linking response and reward is required for optimal learning to occur, it’s difficult to disentangle whether NF training-related differences are due to reinforcement learning based on contingent reward mechanisms or increased/decreased theta power levels per se, or both.I consider that this paragraph forms the cornerstone of this study. Consequently, it must be very clearly formulated. In previous studies, two groups have been considered: an experimental group, in which Fmθ has been up-regulated, and a control sham group.  It is difficult to consider separately the theta increase at Fz lead and the accuracy of contingent feedback. Therefore I ask the authors to take in consideration the following facts to rephrase their research question:     •
In the experimental group, there are:             o
increased  θ power at Fz             o
accurate contingent feedback to ensure optimal learning     •
In the control group,:             o
no optimal learning can be ensured because there is not an accurate contingent feedbackIf authors consider that the previous facts are incorrect, please explain why. 4) lines 113-114: "Another key limitation of existing NF studies, including those targeting Fmθ, is that they modulate neural activity only in one direction (either solely up-modulation or solely down-modulation)." The authors should explain why they do it; that is, their rationale based on EEG activity during attention tasks (Gevins et al., 1997). 5) It must be made very clearly established that this doubt about the placebo sham treatment is only applicable to the Fmθ protocol. However, when using protocols other than this one, the argument that attention could be the one that increased theta activity in Fz has no place. 6) I agree with Reviewer 2 that it is difficult to reach the conclusions the authors reach from the results presented. Perhaps by assembling the results in another way, the authors' conclusion could be accepted, but it is not clear to me in the current form. 7) Not having solved this point (6), it is difficult to form an opinion of the Discussion.

We look forward to receiving your revised manuscript.

Kind regards,

Thalia Fernandez, Ph.D.

Academic Editor

PLOS ONE

Journal Requirements:

JA: Research was sponsored by the Army Research Laboratory and was accomplished under Cooperative Agreement Number W911NF-21-2-0097. The views and conclusions contained in this document are those of the authors and should not be interpreted as representing the official policies, either expressed or implied, of the Army Research Laboratory or the U.S. Government. The U.S. Government is authorized to reproduce and distribute reprints for Government purposes notwithstanding any copyright notation herein.

NB, RR: Research was sponsored by the Army Research Laboratory and was accomplished under Cooperative Agreement Number W911NF-19-2-0106. The views and conclusions contained in this document are those of the authors and should not be interpreted as representing the official policies, either expressed or implied, of the Army Research Laboratory or the U.S. Government. The U.S. Government is authorized to reproduce and distribute reprints for Government purposes notwithstanding any copyright notation herein.

Additional Editor Comments:

1) In lines 54-55 authors postulated that neurofeedback (NF) "is a highly attention-demanding task that must be learned through extended deliberate practice like many skills." This is not in accordance with the fact that in their experiment they only give 5 sessions.

2) lines 82-100: Up-regulation of Fmθ has been used to improve the cognitive performance in healthy young subjects. Up-regulation of Fmθ protocol should not be confused with the protocol to promote REDUCTION in theta power, used by Becerra et al. (2012), in which the subjects were older adults at risk of cognitive decline (evidenced by an excess of theta power for their age, when compared with normative values). Please review the rationale of this last paper.

In this study, a lead was selected in each subject to give the NF. Although in 7 of 14 subjects a frontal lead was selected (in 3 from the experimental group and in 4 from the control group), Fz was never selected; i.e., no subject was down-regulated Fmθ. Therefore, the sentence in lines 98-100 ("Taken together, there is preliminary evidence that either up regulating or down-regulating Fmθ can potentially promote gains in executive performance of some tasks) is not true.

A crucial point in this regard is that these older adults had excess theta at rest, and must be undergoing some compensatory process to correct this abnormality, since their cognitive performance is normal; while the Fmθ protocol of the other studies mentioned has been applied mainly to young subjects for whom the normality / abnormality of their EEG is unknown.

3) lines 105-108: Considering that accurate contingent feedback linking response and reward is required for optimal learning to occur, it’s difficult to disentangle whether NF training-related differences are due to reinforcement learning based on contingent reward mechanisms or increased/decreased theta power levels per se, or both.

I consider that this paragraph forms the cornerstone of this study. Consequently, it must be very clearly formulated. In previous studies, two groups have been considered: an experimental group, in which Fmθ has been up-regulated, and a control sham group. It is difficult to consider separately the theta increase at Fz lead and the accuracy of contingent feedback. Therefore I ask the authors to take in consideration the following facts to rephrase their research question:

• In the experimental group, there are:

o increased θ power at Fz

o accurate contingent feedback to ensure optimal learning

• In the control group,:

o no optimal learning can be ensured because there is not an accurate contingent feedback

If authors consider that the previous facts are incorrect, please explain why.

4) lines 113-114: "Another key limitation of existing NF studies, including those targeting Fm�, is that they modulate neural activity only in one direction (either solely up-modulation or solely down-modulation)." The authors should explain why they do it; that is, their rationale based on EEG activity during attention tasks (Gevins et al., 1997).

5) It must be made very clearly established that this doubt about the placebo sham treatment is only applicable to the Fm protocol. However, when using protocols other than this one, the argument that attention could be the one that increased theta activity in Fz has no place.

6) I agree with Reviewer 2 that it is difficult to reach the conclusions the authors reach from the results presented. Perhaps by assembling the results in another way, the authors' conclusion could be accepted, but it is not clear to me in the current form.

7) Not having solved this point, it is difficult to form an opinion of the Discussion.

Reviewers' comments:

Reviewer's Responses to Questions

**Comments to the Author**

1. Is the manuscript technically sound, and do the data support the conclusions?

Reviewer #1: Partly

Reviewer #2: Yes

Reviewer #3: No

2. Has the statistical analysis been performed appropriately and rigorously? 

Reviewer #1: Yes

Reviewer #2: Yes

Reviewer #3: No

3. Have the authors made all data underlying the findings in their manuscript fully available?

Reviewer #1: Yes

Reviewer #2: Yes

Reviewer #3: Yes

4. Is the manuscript presented in an intelligible fashion and written in standard English?

Reviewer #1: Yes

Reviewer #2: Yes

Reviewer #3: Yes

5. Review Comments to the Author

Reviewer #1: In literature, there is no consensus of how to design aт ultimate neurofeedback study: what EEG parameter to train, what instruction to give, what a comparison neurofeedback option (SHAM?) to use, and what behavioral/neuronal indexes apply to measure the changes.

This study uses a good rational for selection a neurofeedback study: 1) a frontal midline theta rhythm (4-7 Hz) measured at FCz selected as a neurofeedback parameter. 2) No specific instruction was done to modulate the parameter. 3) Reaction time in shooting task (SH) and EEG spectra were used to measure pre-post differences. 4) Two groups with the two types of NF procedures: a) steadily increase theta – INC, and b) alternatively increase/decrease theta - ALT. The study required a lot of hard work of skilled professionals.

However, looking at the results presented in Fig. 4-10, I can’t say that the results are impressive. My impression is that no differences between the groups are found. To prove the opposite (the one that the authors claim in the text), the authors should try to assemble the results in a different and condensed way.

Further, if we accept the results as positive (i.e. the study shows changes between the groups in EEG spectra and behavioral parameters) the Discussion is worth to read.

Reviewer #2: This study encompasses de comparison between two different neurofeedback protocols (increase of Theta and increase/decrease Theta) which aimed at assessing the specificity of NF training overcoming the methodological difficulties of sham and placebo groups.

The authors did a great job, the methods are clear, and the analyses are very good. Just in case, the authors might be interested in the Bayesian LM (i.e. https://pubmed.ncbi.nlm.nih.gov/33486138/)

Here my minor comments:

Intro:

L 93 The authors state that less NF research were down to down training Theta band, however there is a large literature in ADHD which in fact trained down-regulation, although mainly together with up-regulation of the Beta band. This might be mentioned. Here a new reference

Arnold, L. Eugene, Martijn Arns, Justin Barterian, Rachel Bergman, Sarah Black, C. Keith Conners, Shea Connor, u. a. „Double-Blind Placebo-Controlled Randomized Clinical Trial of Neurofeedback for Attention-Deficit/Hyperactivity Disorder With 13 Month Follow-Up“. Journal of the American Academy of Child & Adolescent Psychiatry, 25. August 2020. https://doi.org/10.1016/j.jaac.2020.07.906.

Methods and discussion: Are clear and well written. However, I miss the question regarding specificity and I would recommend performing an additional analysis, which takes into account the block, and session wise learning (i.e. blocke wise / session wise slope on Task performance) and introduce this into the model of the No-go Task. As far as I understood, this was not made. It would add important evidence (or not) of specify of the training which is an important topic.

Reviewer #3: The research question of this study is quite interesting.

However, the proposed methodology does not answer the question. The major problem is that the hypotheses are false, i.e., they are presented as mutually exclusive when, in fact, they are not.

I present some issues that the authors should consider:

INTRODUCTION

- There are several conceptual inaccuracies throughout the introduction. The scope of NF training (i.e., the operant conditioning of brain activity) should be kept in mind in order to avoid mereological fallacies (e.g. lines 67 - 68: NF does NOT train executive control, it trains changes in brain activity, which are not equivalent).

- The authors present a very poor review of previous research regarding fmTheta NF. Only 3 previous studies are shown. More importantly, the 3 studies follow very different rationales:

1. Enriquez-Geppert et al., 2014. They sought to increase fmTheta using NF based on task-induced theta, which is very different to resting-state, task-independent theta activity.

2. Brandmeyer & Delorme, 2020. They also sought to increase fmTheta using NF based on the characteristics of task-specific brain activity, that is, during meditation.

3. Becerra et al., 2012. The description of this article in the introduction is mistaken. Moreover, the rationale behind the NFB treatment has nothing to do with the task-specific fmTheta activity.

- Lines 106-108: “(…)it’s difficult to disentangle whether NF training-related differences are due to reinforcement learning based on contingent reward mechanisms or increased/decreased theta power levels per se, or both.” They are not mutually exclusive. Contingency is a condition for learning to occur. Increases or decreases in theta activity may be a consequence of training (which always involves contingency) or a byproduct of changes in another variable.

- Lines 108-112: The concepts of contingency and learning are inaccurate. The need that the authors mention to implement control conditions with contingent reward is not supported by the previous ideas.

OVERALL PROBLEMS:

- EEG was recorded exclusively during NF training, which may not reflect the EEG activity of the individuals at rest.

- As the authors stated “It is a highly attention-demanding task that must be learned through extended deliberate practice like many skills” (lines 54-55). For this reason, 5 sessions may be insufficient for the actual NF learning to occur.

- Learning curves for each individual should be taken into consideration to decide wether the participant actually learned to modify what the NF protocol intended to modify.

- The change in fmTheta may actually be due to the SH task or the rich and novelty of the VR. The NF protocol could have nothing to do with it.

- An adequate methodology should include, at least, 2 other control groups: One that receives contingent reinforcement for the reduction of fmTheta and another group that receives SHAM feedback. Only then can the research question be truly addressed.

6. PLOS authors have the option to publish the peer review history of their article (what does this mean?). If published, this will include your full peer review and any attached files.

Reviewer #1: No

Reviewer #2: No

Reviewer #3: **Yes: **Mauricio González-López

---

## [Author Response · Author response to Decision Letter 0]

22 Feb 2022

View Letter

Date: Jan 10 2022 10:25AM

To: "Scott E. Kerick" scott.e.kerick.civ@mail.mil

From: "PLOS ONE" plosone@plos.org

Subject: PLOS ONE Decision: Revision required [PONE-D-21-31557]

PONE-D-21-31557

Neural and behavioral adaptations to frontal theta neurofeedback training

PLOS ONE

Dear Dr. Kerick,

Thank you for submitting your manuscript to PLOS ONE. After careful

consideration, we feel that it has merit but does not fully meet PLOS

ONE’s publication criteria as it currently stands. Therefore, we invite

you to submit a revised version of the manuscript that addresses the

points raised during the review process.

1) In lines 54-55 authors postulated that neurofeedback (NF) "is a

highly attention-demanding task that must be learned through extended

deliberate practice like many skills." This is not in accordance with

the fact that in their experiment they only give 5 sessions. This should

be discussed. Even in operant conditioning learning that compromises

survival, such a small number of sessions does not make learning happen;

It is difficult then to believe that it happens when life is not

threatened ...

We agree that the extent of learning afforded in five sessions does not constitute extended deliberate practice. The statement was in fact intended to make this point, that learning to control specific features of one’s brain recordings in a limited number of sessions does not constitute mastery or advanced learning. We moved this statement in the Introduction to the Discussion under “Limitations and Future Directions”. 

2) lines 82-100: Up-regulation of Fmθ has been used to improve the

cognitive performance in healthy young subjects. Up-regulation of Fmθ

protocol should not be confused with the protocol to promote REDUCTION

in theta power, used by Becerra et al. (2012), in which the subjects

were older adults at risk of cognitive decline (evidenced by an excess

of theta power for their age, when compared with normative values).

Please review the rationale of this last paper.

In this study, a lead was selected in each subject to give the NF.

Although in 7 of 14 subjects a frontal lead was selected (in 3 from the

experimental group and in 4 from the control group), Fz was never

selected; i.e., no subject was down-regulated Fmθ. Therefore, the

sentence in lines 98-100 ("Taken together, there is preliminary evidence

that either up regulating or down-regulating Fmθ can potentially promote

gains in executive performance of some tasks) is not true.

A crucial point in this regard is that these older adults had excess

theta at rest, and must be undergoing some compensatory process to

correct this abnormality, since their cognitive performance is normal;

while the Fmθ protocol of the other studies mentioned has been applied

mainly to young subjects for whom the normality / abnormality of their

EEG is unknown.

We agree with this important criticism. Becerra et al., and much of the literature on Fmθ NF training, has been done on patient populations (e.g., ADHD, elderly) which exhibit excess frontal theta thought to be disruptive to cognitive performance. We revised our referencing of this study by more detailed review of this and other NF studies originally cited, stating that Fmθ was not modulated in Becerra et al., and describing the differences between studies on healthy young adults and elderly/patient. We then focus on the fact that no previous researchers have considered alternating Fmθ in a manner that is more consistent with frontal theta dynamics (increases/decreases) during executive control tasks (activation/inhibition); see revised section “Theta: A promising neural target for training executive control”.

3) lines 105-108: Considering that accurate contingent feedback linking

response and reward is required for optimal learning to occur, it’s

difficult to disentangle whether NF training-related differences are due

to reinforcement learning based on contingent reward mechanisms or

increased/decreased theta power levels per se, or both.

I consider that this paragraph forms the cornerstone of this study.

Consequently, it must be very clearly formulated. In previous studies,

two groups have been considered: an experimental group, in which Fmθ has

been up-regulated, and a control sham group. It is difficult to

consider separately the theta increase at Fz lead and the accuracy of

contingent feedback. Therefore I ask the authors to take in

consideration the following facts to rephrase their research question:

 • In the experimental group, there are:

 o increased θ power at Fz

 o accurate contingent feedback to ensure optimal learning

 • In the control group,:

 o no optimal learning can be ensured because there is not an accurate contingent feedback

If authors consider that the previous facts are incorrect, please

explain why.

We agree with these comments and facts. Accurate contingent feedback is essential for optimal learning to occur. However, just because accurate contingent feedback is provided in NF training groups but not sham control groups, does not necessarily mean that greater increases in Fm� in the NF group is clear and unambiguous evidence that it is because of this coupling between accurate contingent feedback and increased theta power. It cannot be assumed, alternative hypotheses should be rejected. The increased theta could alternatively be a function of different levels of motivation, attentional engagement, task difficulty, strategies, or any of a number of other possibilities. We have attempted to more clearly make this point in the revision under the section “Unresolved Issues and Gaps”.

4) lines 113-114: "Another key limitation of existing NF studies,

including those targeting Fmθ, is that they modulate neural activity

only in one direction (either solely up-modulation or solely

down-modulation)." The authors should explain why they do it; that is,

their rationale based on EEG activity during attention tasks (Gevins et

al., 1997).

The main point here is that, although higher levels of frontal theta characterize attentional engagement in attention tasks, it is not stationary. Rather, as event-related spectral perturbation studies indicate, theta is phasically modulated across trials of executive tasks. Therefore, it is tenable to conjecture that alternating the power of frontal theta over the course of training may be a more task-specific training protocol than exclusively increasing it to that which is exhibited during task performance (task specificity). An analogy might be training a sprinter (50 m) by having him/her run long distances (5 K).

In lines 76-83, under “Theta: A promising neural target for training executive control”, it was our intent to provide a rationale for why NF training is designed to increase Fm� power:–

“Extensive research has shown that EEG-theta synchronization is central to executive control (Cavanagh et al., 2009; Luu et al., 2004; Mizuhara & Yamaguchi, 2007; Polania et al., 2012; Womelsdorf, Johnston, Vinck, & Everling, 2010; Womelsdorf, Vinck, Leung, & Everling, 2010). Theta synchronization functions to enable long-distance communication in the brain, dynamically linking subcortical and cortical brain networks associated with integrating and coordinating perceptual, cognitive, and motor control functions.”

However, our main point in lines 113-114 was to emphasize the dynamic and nonstationary nature of phasic theta changes during attention tasks: 

“However, to support adaptive behavior in real-world situations, neural activity must exhibit dynamic variability such that activation and inhibition must be coordinated in response to varied internal/external demands.”

We agree that Gevins et al. (1997) is a relevant citation to include in making this point that frontal theta increases in response to executive task demands, and therefore provides further justification for up-modulating Fm� in NF training. We have included this reference in the above paragraph stating that EEG-theta synchronization is central to executive control.

5) It must be made very clearly established that this doubt about the

placebo sham treatment is only applicable to the Fmθ protocol. However,

when using protocols other than this one, the argument that attention

could be the one that increased theta activity in Fz has no place.

We agree, that this issue is specific to Fmθ. We believe that we make this clear in the first few sentences under “Unresolved Issues and Gaps” 

“Representing another key issue, analogous to the above on sham control issues, the Fmθ activation targeted by NF is often confounded with the neural functions required to modulate brain responses in NF protocols. Specifically, NF is an attention-demanding task requiring the activation of frontal executive control networks during training irrespective of the frequency or region targeted by feedback (Gruzelier, 2014; Haller et al., 2013; Emmert et al., 2016). Since Fmθ is implicated in such executive control processes, observed Fmθ increases during NF training may be erroneously attributed to effective NF training (i.e. reinforcement learning based on accurate contingent feedback) when such increases may merely indicate sustained deployment of attention and/or cognitive effort.” 

6) I agree with Reviewer 2 that it is difficult to reach the conclusions

the authors reach from the results presented. Perhaps by assembling the

results in another way, the authors' conclusion could be accepted, but

it is not clear to me in the current form.

We are not sure how to reassemble the results in a more clear or refined way. We believe it is important to first show how the different training protocols resulted in different Fm� changes between groups over blocks and sessions. Establishing that the ALT group exhibited higher levels of Fm� over blocks and sessions suggests that accurate contingent reward cannot account for these findings. If feedback enabled learning to occur, then the ALT group should have been able to downregulate their Fm� levels more during down vs up-regulation blocks, and they should have generated lower Fm� than the INC group across both blocks and sessions. Therefore, we conclude that some other mechanisms must be accountable for this finding (e.g., differences in cognitive demands, fatigue, or other).

Then, to test how the training affected Fm� and performance during the Go-NoGo task, we follow with neural and behavioral results. The findings are a bit difficult to clearly comprehend, as no differences were observed for Fm�, while behavioral results differed depending on the particular measure. However, short of leaving out specific results for convenience or brevity, it’s hard to portray the complete picture. We hope that by revising our Introduction, the questions and hypotheses are more clear, thereby making the Results and Discussion/Conclusions more clear.

7) Not having solved this point (6), it is difficult to form an opinion

of the Discussion.

Please see reply to (6) above.

5. Review Comments to the Author

Reviewer #1: In literature, there is no consensus of how to design aт

ultimate neurofeedback study: what EEG parameter to train, what

instruction to give, what a comparison neurofeedback option (SHAM?) to

use, and what behavioral/neuronal indexes apply to measure the changes.

We agree with this comment.

This study uses a good rational for selection a neurofeedback study: 1)

a frontal midline theta rhythm (4-7 Hz) measured at FCz selected as a

neurofeedback parameter. 2) No specific instruction was done to modulate

the parameter. 3) Reaction time in shooting task (SH) and EEG spectra

were used to measure pre-post differences. 4) Two groups with the two

types of NF procedures: a) steadily increase theta – INC, and b)

alternatively increase/decrease theta - ALT. The study required a lot of

hard work of skilled professionals.

However, looking at the results presented in Fig. 4-10, I can’t say that

the results are impressive. My impression is that no differences between

the groups are found. To prove the opposite (the one that the authors

claim in the text), the authors should try to assemble the results in a

different and condensed way.

Based on our statistical analyses, we did observe significant group differences in Fm� over blocks and sessions of NF training, and behavioral measures in the Go-NoGo task, but we did not observe differences in Fm� over sessions in the Go-NoGo task. These findings are difficult to clearly interpret, and may be considered unimpressive. However, we argue that our results contradict previous research, and that reporting these results is relevant to the corpus of NF literature in stimulating new research that should be designed to reject alternative hypotheses (rather than support existing ones) that training-related increases in Fm� underlie executive task improvement. In our view, based on our results alternative hypotheses remain tenable, and creative implementation of alternative control groups beyond just sham control should be encouraged to advance our understanding of NF training. 

Further, if we accept the results as positive (i.e. the study shows

changes between the groups in EEG spectra and behavioral parameters) the

Discussion is worth to read.

We hope that by revising our Introduction, the questions and hypotheses are more clear, thereby making the Results and Discussion/Conclusions more clear.

Reviewer #2: This study encompasses de comparison between two different

neurofeedback protocols (increase of Theta and increase/decrease Theta)

which aimed at assessing the specificity of NF training overcoming the

methodological difficulties of sham and placebo groups.

The authors did a great job, the methods are clear, and the analyses are

very good. Just in case, the authors might be interested in the Bayesian

LM (i.e. https://pubmed.ncbi.nlm.nih.gov/33486138/)

We thank the reviewer for the suggestion. We view Konicar et al., (2021) work to be an interesting approach to analyzing neurofeedback data. Given our data distribution we feel that our analyses (in this case) are best supported by using restricted maximum likelihood—as part of the multi-level modeling approach (maximum likelihood is a core component of Bayesian analyses). 

Here my minor comments:

Intro:

L 93 The authors state that less NF research were down to down training

Theta band, however there is a large literature in ADHD which in fact

trained down-regulation, although mainly together with up-regulation of

the Beta band. This might be mentioned. Here a new reference

Arnold, L. Eugene, Martijn Arns, Justin Barterian, Rachel Bergman, Sarah

Black, C. Keith Conners, Shea Connor, u. a. „Double-Blind

Placebo-Controlled Randomized Clinical Trial of Neurofeedback for

Attention-Deficit/Hyperactivity Disorder With 13 Month Follow-Up“.

Journal of the American Academy of Child & Adolescent Psychiatry, 25.

August 2020. https://doi.org/10.1016/j.jaac.2020.07.906.

We thank the reviewer for the suggestion and reference. We have incorporated a few sentences acknowledging that downregulation of frontal theta is frequently implemented in research on elderly and patient groups and included this reference, among a few more.

Methods and discussion: Are clear and well written. However, I miss the

question regarding specificity and I would recommend performing an

additional analysis, which takes into account the block, and session

wise learning (i.e. blocke wise / session wise slope on Task

performance) and introduce this into the model of the No-go Task. As far

as I understood, this was not made. It would add important evidence (or

not) of specify of the training which is an important topic.

In general we agree with the reviewer. However, there are a few reasons why we did not analyze the data at the block level for the Go-NoGo task. First, the block-level data consists of only 9 no-go trials (10% of 90), making analyses of Fm� to friendly targets and errors of commission unstable. Second, we are concerned about overfitting a model with additional factors (Group, Session, Condition, Block) given only 30 subjects. Third, for the first two reasons, we did not partition the data at the block level, and doing so now would take more time than we have to submit a revision. 

Reviewer #3: The research question of this study is quite interesting.

However, the proposed methodology does not answer the question. The

major problem is that the hypotheses are false, i.e., they are presented

as mutually exclusive when, in fact, they are not.

I present some issues that the authors should consider:

INTRODUCTION

- There are several conceptual inaccuracies throughout the introduction.

The scope of NF training (i.e., the operant conditioning of brain

activity) should be kept in mind in order to avoid mereological

fallacies (e.g. lines 67 - 68: NF does NOT train executive control, it

trains changes in brain activity, which are not equivalent).

We agree with this important criticism. We have revised this sentence: “As such, using NF to train modulation of neural functions related to executive control”, and throughout the Introduction. We agree with Rogala et al. (2016), who claimed that “NFB protocols can be measured by unambiguous changes in EEG activity and by changes in the targeted cognitive function”. We have incorporated Rogala et al. in paragraph two under the sub-heading: “Theta: A promising neural target for training executive control”.

- The authors present a very poor review of previous research regarding

fmTheta NF. Only 3 previous studies are shown. More importantly, the 3

studies follow very different rationales:

We realize that there is a wealth of research on NF training among various subject populations (children/adult/elderly, clinical/normal) targeting various frequency bands, brain regions, and training goals. However, there is a paucity of research on Fm� NF training on healthy young adults for cognitive performance enhancement. Many studies have investigated ratios of frontal theta with other frequency bands (e.g., alpha, beta); however, training-related changes of ratios are difficult to interpret because it is not known whether the changes observed are due to theta or some other band, or both, unless absolute values are also included. Further, elderly and clinical populations generally exhibit overactive frontal theta, which is why down-regulation goals of training are implemented, but these populations differ from our young healthy subjects and our training goals are different. For this reason, we necessarily limited our literature review to studies specifically designed to investigate purely Fm� NF training on healthy young adults for cognitive performance enhancement, which required us to summarize studies with different rationales. We have incorporated more detailed review of the most highly relevant studies to better illuminate consistencies/inconsistencies, and outstanding issues. We based most of our review of relevant literature on a recent review by Rogala et al. (2016), which we have inserted in paragraph two under “Theta: A promising neural target for training executive control”:

In their recent meta-analytic review of the NF literature, Rogala et al. (2016) concluded that, “the validity of EEG-NFB protocols can be measured by unambiguous changes in EEG activity and by changes in the targeted cognitive function. Unfortunately, most of the work conducted in the EEG-NFB field has failed to satisfy the unambiguity criterion for both of the variables and the field itself has shown a big tolerance for violations of scientific methodology” (Rogala et al., 2016, p. 2). They went on to say that “a rigorous scientific approach to EEG-NFB is rare, and experiments performed on healthy participants to study the effectiveness and/or mechanism of training are very limited” (Rogala et al., 2016, p. 8). Based on their review, only five theta NF studies met these criteria (Becerra et al., 2011; Enriquez-Geppert et al., 2013; Enriquez-Geppert et al., 2014; Reis et al., 2015; Wang & Hsieh et al., 2013). Four of five studies implemented theta up-regulation, and all five revsealed significant EEG and behavioral changes in NF vs. control groups (more recently, we also included Brandmeyer & Delorme, 2020).

1. Enriquez-Geppert et al., 2014. They sought to increase fmTheta using

NF based on task-induced theta, which is very different to

resting-state, task-independent theta activity.

Enriquez-Geppert et al. (2014) did use executive control tasks to determine individual peak theta frequencies for training; however, the threshold levels of resting baseline Fm� preceding NF training was used as reference thresholds for each training session: “the raw power value of each 2 s segment for the individualized fm-theta frequency was compared to the baseline power of the same frequency… power change [was] relative to the start-baseline of the specific NF session” (p. 5). Further the other studies we reviewed also used pre-training rest baselines as threshold for training (Brandmeyer & Delorme, 2020; Wang & Hsieh, 2013). We adopted this approach in our study in the first session (training threshold based on preceding resting state Fm�), but after the first session, in subsequent sessions, we used the mean Fm� from the preceding session as the threshold. In this way, our approach was actually based on Fm� generated during NF training sessions, after the first session. We believe this approach would make training more task-dependent. Further, we know from previous research that frontal theta phasically increases in response to targets and non-targets in Go-NoGo shooting tasks (Kerick, Hatfield, & Allender, 2007; https://www.ingentaconnect.com/content/asma/asem/2007/00000078/A00105s1/art00024), so our training was not based on resting state theta.

2. Brandmeyer & Delorme, 2020. They also sought to increase fmTheta

using NF based on the characteristics of task-specific brain activity,

that is, during meditation.

Brandmeyer & Delorme (2020) did have their subjects modulate Fm� while simultaneously monitoring breathing cycles, which actually was a form of dual-task training. This does complicate the interpretation to some extent. However, given the relative paucity of Fm� neurofeedback research on healthy individuals designed to improve executive function, we feel it is important to include this study in our summary of the relevant literature. To point out the different characteristics of this study, we have elaborated our review of it in the Introduction. As mentioned above, Brandmeyer & Delorme also based their Fm� training thresholds on resting state theta preceding training in the first block if the first session, and subsequently the threshold was based on Fm� from the previous block: “… the 1 min preparatory baseline period which preceded the first feedback session which allowed the program to calculate an acceptable theta dynamic range for the onset of the first neurofeedback block. Dynamic range at the end of each of the 5 min block, was used as a starting point for the following 5 min Blocks” (p. 6). Our thresholds for training were implemented similar to this: thresholds were based on resting theta in first session, but then was based on changes over training sessions.

3. Becerra et al., 2012. The description of this article in the

introduction is mistaken. Moreover, the rationale behind the NFB

treatment has nothing to do with the task-specific fmTheta activity.

We agree that the description of Becerra et al. was not accurately portrayed and should not have been discussed in the context of decreasing Fm� in healthy young adults. In that study, elderly subjects with abnormally high levels of theta across various brain regions were tested. Accordingly, we have revised the manuscript under the sub-heading “Theta: A promising neural target for training executive control”. 

In general, we argue that NF may be task-specific or general, depending on the goals of the training, targeted neural features, and training schedule (Papo, 2019). Specific to Becerra et al., we agree that training was not designed with task-specificity in mind: “The goal of this work was to explore the effectiveness of a NFB protocol in reducing theta EEG activity in normal elderly subjects who present abnormally high theta… with the aim of reducing the probability of posterior cognitive decline” (p. 2).

- Lines 106-108: “(…)it’s difficult to disentangle whether NF

training-related differences are due to reinforcement learning based on

contingent reward mechanisms or increased/decreased theta power levels

per se, or both.” They are not mutually exclusive. Contingency is a

condition for learning to occur. Increases or decreases in theta

activity may be a consequence of training (which always involves

contingency) or a byproduct of changes in another variable.

We agree that contingency is a condition for learning and the point we were trying to make was not clear. That is, it cannot be ruled out from previous research whether differences in Fm� between real and sham groups are due to contingent reward mechanisms. Alternative mechanisms are also plausible (differences in motivation/engagement, attentional effort, boredom, fatigue, etc.), especially considering that sham control subjects have been shown to report knowledge of lack of contingency (Brandmeyer & Delorme, 2020). Further, in many studies the sham group also showed increases in pre-post task performance, suggesting that behavioral changes may be related to some non-specific factors unrelated to contingent reward of NF training. Thus, we argue that research is needed which consists of control and experimental groups each receiving contingent reward, but differing in the direction of targeted Fm� modulation. In particular, previous NF research consists of training prolonged sustained Fm� activation (5-30 min), whereas in the performance of executive tasks, frontal theta phasically increases and decreases over the course of trials. In this regard, we agree with Papo (2019) that NF training should consider dynamics-to-function mapping (although our study was designed and conducted before Papo (2019)). We have attempted to present a more clear argument in our revised Introduction, particularly under the subheading “Unresolved Issues and Gaps”. 

- Lines 108-112: The concepts of contingency and learning are

inaccurate. The need that the authors mention to implement control

conditions with contingent reward is not supported by the previous ideas.

We disagree with this criticism. It is possible to disentangle the question of whether changes in Fm� over training sessions are associated with reinforcement learning based on contingent reward mechanisms or based on differences in other potential cognitive factors such as motivation/engagement, attentional effort, boredom, fatigue, etc. We attempted to make this argument more clearly in the revised Introduction.

OVERALL PROBLEMS:

- EEG was recorded exclusively during NF training, which may not reflect

the EEG activity of the individuals at rest.

To evaluate training-related changes in Fm�, we evaluated Fm� relative to previous training power levels of Fm�, except in the first session, for which we used resting Fm� as the threshold power level. 

- As the authors stated “It is a highly attention-demanding task that

must be learned through extended deliberate practice like many skills”

(lines 54-55). For this reason, 5 sessions may be insufficient for the

actual NF learning to occur.

We agree that five sessions is not sufficient extended deliberate practice. This point was made to elucidate a general issue of NF training, which is that most studies (including ours) do not train individuals over sufficient time to observe robust and lasting learning effects. We have moved this point from the Introduction to the Discussion under the sub-heading “Limitations and Future Directions”.

- Learning curves for each individual should be taken into consideration

to decide wether the participant actually learned to modify what the NF

protocol intended to modify.

Because a multi-level modeling approach was used individual variance (i.e. changes at the individual level) were accounted for. As mentioned at the end of the statistical analyses section a responder versus non-responder (i.e. whether a participant successfully altered their theta over time) analysis indicated that differences were not observed between responders and non-responders: 

“A responder-only analysis was also conducted but not reported due to few differences in the results compared to using the full data. For the analysis, participants were considered responders if either session or moderation could predict their Fm� power during NF training using univariate regression analyses. Using these criteria 18 participants were considered responders and 12 were non-responders. Again, due to limited impact on the results the full data were used”.

- The change in fmTheta may actually be due to the SH task or the rich

and novelty of the VR. The NF protocol could have nothing to do with it.

No differences in Fm� were observed in the SH task, only in the NF training were Fm� differences observed. Further, NF training occurred before SH in each session, except the first, so it’s unlikely novelty effects persisted across all sessions, or that the SH task in prior sessions affected the NF training in subsequent sessions. 

- An adequate methodology should include, at least, 2 other control

groups: One that receives contingent reinforcement for the reduction of

fmTheta and another group that receives SHAM feedback. Only then can the

research question be truly addressed.

We agree with this statement and have included it in the section “Limitations and Future Directions”. Ideally we would have included a sham control group and a down-modulation group, however, practical considerations need to be considered as well (time, cost, subject and experimentor commitments, etc.).

---

## [Decision Letter · Decision Letter 1]

3 Jun 2022

PONE-D-21-31557R1Neural and behavioral adaptations to frontal theta neurofeedback trainingPLOS ONE

Dear Dr. Kerick,

Thank you for submitting your manuscript to PLOS ONE. After careful consideration, we feel that it has merit but does not fully meet PLOS ONE’s publication criteria as it currently stands. Therefore, we invite you to submit a revised version of the manuscript that addresses the points raised during the review process.

The reviewers and I think that the authors have improved the manuscript substantially, meanly the Introduction. However, the current version of the manuscript does not fully ready for publication yet.

Although the response letter addressed the reviewers comments, the manuscript still does not reflect what was written in the response letter.  I suggest that the authors indicate precisely the lines of the document in which these changes are reflected.

The most serious problem of the article is the very small number of sessions. It is a pity that only 5 sessions have been applied, because in my opinion they are insufficient to reach a conclusion. Please review Alatorre-Cruz et al. (2022). In the penultimate paragraph of their discussion ("A strength of our study is that 30 sessions of treatment were applied...") reference is made to John Garcia's classic work on Pavlovian flavor conditioning. Even in extreme cases of survival, it is sometimes not possible to establish learning with a few training sessions. The experiment carried out in this paper is not related to survival. Therefore, although the exposed theory is interesting, it should be handled as a proof of concept without trying to reach conclusions based on the experiment. The Discussion must be much more critical regarding this weakness of the study.

There are attached some suggestions that could help improve the introduction.

We look forward to receiving your revised manuscript.

Kind regards,

Thalia Fernandez, Ph.D.

Academic Editor

PLOS ONE

Additional Editor Comments:

The reviewers and I think that the authors have improved the manuscript substantially, meanly the Introduction. However, the current version of the manuscript does not fully ready for publication yet.

Although the response letter addressed the reviewers comments, the manuscript still does not reflect what was written in the response letter. I suggest that the authors indicate precisely the lines of the document in which these changes are reflected.

The most serious problem of the article is the very small number of sessions. It is a pity that only 5 sessions have been applied, because in my opinion they are insufficient to reach a conclusion. Please review Alatorre-Cruz et al. (2022). In the penultimate paragraph of their discussion ("A strength of our study is that 30 sessions of treatment were applied...") reference is made to John Garcia's classic work on Pavlovian flavor conditioning. Even in extreme cases of survival, it is sometimes not possible to establish learning with a few training sessions. The experiment carried out in this paper is not related to survival. Therefore, although the exposed theory is interesting, it should be handled as a proof of concept without trying to reach conclusions based on the experiment. The Discussion must be much more critical regarding this weakness of the study.

Here are some suggestions that could help improve the introduction:

Introduction

1.- Neurofeedback (NF) is a psychophysiological training, therefore there must be scientific evidence of a clear relationship between the particular neurophysiological marker targeted by neurofeedback and the behavioral or cognitive process being studied. This is not clear in the description of NF the authors give.

2.- line 51 & 55

"involves defining: i) the general goal;..."

"Depending on one’s goals, training..."

researcher's goal regarding what? conduct? EEG? another thing?

3.- line 87-90

"For example, Enriquez-Geppert et al., (2014) found that N-back and task-switching performance improved pre-post training in both NF and sham groups, but to a greater extent in the NF group; however, performance on stop-signal and Stroop tasks did not change as a function of real or sham training."

If both groups improved their performance, this improvement is not an exclusive consequence of operant conditioning. What happened in the Sham group could be a consequence of other factors (see Alatorre et al 2022).

4.- lines 94-100

"Less research has used NF to promote decreases in theta power. In one study using this protocol, older participants instructed to sustain decreased levels of theta power, presented as the lead where theta/alpha ratios were most abnormally high (in 6 of 14 total subjects a frontal lead was identified), the NF group exhibited higher performance gains on verbal comprehension and verbal IQ relative to a random feedback group, but no differences were observed on executive IQ (Becerra et al., 2011). Taken together, there is preliminary evidence that either up-regulating or down-regulating Fm� can potentially promote gains in executive performance of some tasks."

I invite the authors to review the article by Alatorre-Cruz et al. (2022), recently published by the same research group as Becerra et al. (2012). I consider that it is very important to highlight that the healthy older adults in both articles had an electroencephalographic risk of developing cognitive impairment (MCI or dementia); that risk consisted of having an abnormally high value of absolute power theta. This marks a big difference between the two types of participants that are pointed out: some are healthy young adults (it is not known if theta activity is normal or not) and others are clinically healthy older adults with abnormal theta activity. In the former, the objective is to improve performance but in the latter, the objective is to prevent cognitive deterioration.

The authors talk about the theta/alpha ratio, but this seems to be an error because it is not directly related to the study' rationale and is not mentioned by Alatorre-Cruz et al., who said they are replicating the Becerra experiment and evaluating a 1-year follow-up.

5.- lines 105-108

"Considering that accurate contingent feedback linking response and reward is required for optimal learning to occur (Sterman & Egner, 2006), it’s difficult to disentangle whether NF training-related differences are due to reinforcement learning based on contingent reward mechanisms or increased/decreased theta power levels per se, or both. Contingent reward enables the accurate coupling of top-down goals and bottom-up stimulus processing (error monitoring and control) that may account for the learning differences observed in previous sham control studies. However, a need exists to implement control conditions in which accurate contingent reward is also inherent in the training and consistent with experimental NF training conditions. However, a need exists to implement control conditions in which accurate contingent reward is also inherent in the training and consistent with experimental NF training conditions."

I exhort to the authors to better explain the ideas because I do not understand. The conclusions that are given throughout the paragraph are not derived from what has been said previously.

6.- lines 113- 117

"Another key limitation of existing NF studies, including those targeting Fm�, is that they modulate neural activity only in one direction (either solely up-modulation or solely down-modulation). However, to support adaptive behavior in real-world situations, neural activity must exhibit dynamic variability such that activation and inhibition must be coordinated in response to varied internal/external demands. "

References are required

7.- lines 123-127

" In these views, it is not sustained increases or decreases in theta (or any other neural process) that are most optimal for adaptive cognitive performance. Rather, dynamically shifting between increasing and decreasing activity (up and down modulation) might promote more efficient neural adaptations in support of superior executive control performance."

Is this the authors' interpretation of what was previously said or are there experiments that prove it?

8.- lines 128-131

Representing another key issue, analogous to the above on sham control issues, the Fm� activation targeted by NF is often confounded with the neural functions required to modulate brain responses in NF protocols. Specifically, NF is an attention-demanding task requiring the activation of frontal executive control networks during training irrespective of the frequency or region targeted by feedback

It is not clear "the Fm� activation targeted by NF is often confounded with the neural functions required to modulate brain responses in NF protocols."

9.- line 141:

"This possibility obscures extant findings on Fm��modulation, such that observed effects of Fm��modulation on the brain and performance may not be driven by NF per se (i.e. reinforcement learning) but instead by differences in cognitive control strategies and effort levels when attempting NF (i.e. theta activation)"

When authors say "(i.e. reinforcement learning)" are they referring to "operant conditioning"? Terms must be used precisely. Operant conditioning can be done using reinforcers or punishments.

10.- lines 180-189

" We tested two competing hypotheses regarding the effects of NF training on concomitant levels of Fm��power. If Fm��changes are due to reinforcement learning afforded by accurate contingent feedback, then the INC group should exhibit greater increases within and across training sessions. If Fm��changes are due to greater executive task demands associated with task-switching of top-down goals, then the ALT group should exhibit greater increases within and across training sessions. Assuming that Fm��is effectively shaped by reinforcement learning in both groups, we also hypothesized that the ALT group will improve at a greater rate on the Go-NoGo shooting task over sessions. Furthermore, we hypothesized higher Fm��power in the ALT group during performance of the Go-NoGo task over sessions, suggesting more refined task specificity and context dependent adaptations, or more specific dynamics-to-function mapping, in the ALT group."

Please try to explain more clearly the rationale of these hypotheses.

Reviewers' comments:

Reviewer's Responses to Questions

**Comments to the Author**

1. If the authors have adequately addressed your comments raised in a previous round of review and you feel that this manuscript is now acceptable for publication, you may indicate that here to bypass the “Comments to the Author” section, enter your conflict of interest statement in the “Confidential to Editor” section, and submit your "Accept" recommendation.

Reviewer #1: All comments have been addressed

Reviewer #3: (No Response)

2. Is the manuscript technically sound, and do the data support the conclusions?

Reviewer #1: Yes

Reviewer #3: Partly

3. Has the statistical analysis been performed appropriately and rigorously? 

Reviewer #1: Yes

Reviewer #3: No

4. Have the authors made all data underlying the findings in their manuscript fully available?

Reviewer #1: Yes

Reviewer #3: Yes

5. Is the manuscript presented in an intelligible fashion and written in standard English?

Reviewer #1: Yes

Reviewer #3: Yes

6. Review Comments to the Author

Reviewer #1: I think that the authors have improved the manuscript substantially. I also think that the field needs to see the results of this important study.

Reviewer #3: The authors have addressed in an adequate manner the reviewers comments in their response letter. The aim of the study seems more clear to me now.

However, the manuscript still does not reflect what was written in the response letter to the reviewers. The issue of mereological fallacy is still found throughout the manuscript (e.g. lines 67 & 70).

Please make sure that the whole manuscript is coherent with the answers that you have provided to the reviewers.

Moreover, I encourage the author to include the results of the comparison between responders and non-responders. Even though you have found no differences, this is an important issue, because it is directñy related to your research question.

7. PLOS authors have the option to publish the peer review history of their article (what does this mean?). If published, this will include your full peer review and any attached files.

Reviewer #1: No

Reviewer #3: **Yes: **Mauricio González-López

---

## [Author Response · Author response to Decision Letter 1]

17 Oct 2022

See uploaded document "Responses to R1 comments 2.docx"

---

## [Decision Letter · Decision Letter 2]

15 Nov 2022

PONE-D-21-31557R2Neural and behavioral adaptations to frontal theta neurofeedback training

PLOS ONE

Dear Dr. Kerick,

Thank you for submitting your manuscript to PLOS ONE. After careful consideration, we feel that it has merit but does not fully meet PLOS ONE’s publication criteria as it currently stands. Therefore, we invite you to submit a revised version of the manuscript that addresses the points raised during the review process.

Taking into consideration Reviewer 3's comments, the authors could discuss as a weakness of their project the fact that the hypotheses they considered are not mutually exclusive. Being the current document a proof of concept, mainly due to the small sample size, it is reasonable to assume that performing the experiment will overcome this weakness. Please consider making it explicit in the title that this is a proof of concept.

We look forward to receiving your revised manuscript.

Kind regards,

Thalia Fernandez, Ph.D.

Academic Editor

PLOS ONE

Journal Requirements:

Reviewers' comments:

Reviewer's Responses to Questions

**Comments to the Author**

1. If the authors have adequately addressed your comments raised in a previous round of review and you feel that this manuscript is now acceptable for publication, you may indicate that here to bypass the “Comments to the Author” section, enter your conflict of interest statement in the “Confidential to Editor” section, and submit your "Accept" recommendation.

Reviewer #3: All comments have been addressed

2. Is the manuscript technically sound, and do the data support the conclusions?

Reviewer #3: Partly

3. Has the statistical analysis been performed appropriately and rigorously? 

Reviewer #3: Yes

4. Have the authors made all data underlying the findings in their manuscript fully available?

Reviewer #3: Yes

5. Is the manuscript presented in an intelligible fashion and written in standard English?

Reviewer #3: Yes

6. Review Comments to the Author

Reviewer #3: This paper addresses the question of whether the previously reported increases in FM-theta activity as a consequence of neurofeedback training could be actually attributed to the actual feedback procedure (i.e., administering a contingent reward to ongoing changes in fm-theta increases during training) or if they could be a consequence of non-specific executive control processes involved in NFB training.

In order to answer the research question, the authors aimed to test two competing hypotheses regarding the effects of NFB training on fm-theta power:

1. If Fm-theta changes are due to self-regulation afforded by accurate contingent feedback, then the INC group should exhibit greater increases within and across training sessions, and

2. If, on the other hand, Fm-theta increases more in the ALT group or no differences are observed between groups, then self-regulation based on accurate contingent reward mechanisms underlying reinforcement learning would be challenged.

If this were taken to be true, then it would involve that actual operant learning as a consequence of providing contingent feedback and the non-specific executive control processes are mutually exclusive, which is not the case. The increases in fm-theta could also be a result of a combination of these factors or they could also be attributable to another variable altogether (for example expectancy, i.e., placebo effect). Moreover, the ALT group could actually get better at increasing fm-theta activity because contingent feedback provides them with better contextual cues to achieve discrimination of their own neurophysiological state.

Also, the authors should keep in mind that an actual increase in behavior is necessary for a stimulus to be a reinforcer, contingency alone is not sufficient to establish the functionality of a stimulus as a reward (reinforcer). That is, for the feedback to actually be a reinforcer, one must see an increase in the behavior that is pretended to be reinforced.

Since the experimental design assumes the hypotheses to be mutually exclusive, the research method proposed by the authors does not adequately address their research question.

7. PLOS authors have the option to publish the peer review history of their article (what does this mean?). If published, this will include your full peer review and any attached files.

Reviewer #3: **Yes: **Mauricio González-López

---

## [Author Response · Author response to Decision Letter 2]

11 Jan 2023

Reviewer #3: This paper addresses the question of whether the previously reported increases in FM-theta activity as a consequence of neurofeedback training could be actually attributed to the actual feedback procedure (i.e., administering a contingent reward to ongoing changes in fm-theta increases during training) or if they could be a consequence of non-specific executive control processes involved in NFB training.

In order to answer the research question, the authors aimed to test two competing hypotheses regarding the effects of NFB training on fm-theta power:

1. If Fm-theta changes are due to self-regulation afforded by accurate contingent feedback, then the INC group should exhibit greater increases within and across training sessions, and

2. If, on the other hand, Fm-theta increases more in the ALT group or no differences are observed between groups, then self-regulation based on accurate contingent reward mechanisms underlying reinforcement learning would be challenged.

If this were taken to be true, then it would involve that actual operant learning as a consequence of providing contingent feedback and the non-specific executive control processes are mutually exclusive, which is not the case. The increases in fm-theta could also be a result of a combination of these factors or they could also be attributable to another variable altogether (for example expectancy, i.e., placebo effect). Moreover, the ALT group could actually get better at increasing fm-theta activity because contingent feedback provides them with better contextual cues to achieve discrimination of their own neurophysiological state.

Response: We agree with these points and have incorporated them into the revised manuscript. 

[Lines 304-307] After stating hypotheses, we added the following sentence: “These hypotheses assume that self-regulation processes based on operant conditioning are differentiable from non-specific executive control processes. Further, they assume an adequate stimulation schedule to enable sufficient reinforcement learning”

[Lines 736-742] To reinforce these points in the Discussion, we also revised the 2nd paragraph in the Discussion: “In sham control studies, it is assumed that reinforcement learning afforded by accurate contingent feedback in the real NF training group enables learning of self-regulation, while the lack of accurate contingent feedback in sham controls precludes it. Further, it assumes that self-regulation processes afforded by accurate contingent feedback and non-specific executive control processes such as attention and effort are mutually exclusive.”

[Lines 981-985] To further reinforce these points in the Conclusions, we added: “From a complex systems perspective, assumptions about differentiation (i.e., integration/segregation) of multiple concurrent executive functions (self-monitoring, attention, conflict monitoring, action control, decision making, memory, etc). should be more carefully considered and challenged whether sham control or alternative control conditions are employed in NF training research.”

Also, the authors should keep in mind that an actual increase in behavior is necessary for a stimulus to be a reinforcer, contingency alone is not sufficient to establish the functionality of a stimulus as a reward (reinforcer). That is, for the feedback to actually be a reinforcer, one must see an increase in the behavior that is pretended to be reinforced.

Response: We agree with this point and have incorporated this into the revised manuscript. [Lines 304-307]

Since the experimental design assumes the hypotheses to be mutually exclusive, the research method proposed by the authors does not adequately address their research question

Response: We have emphasized this point in the above revisions. We believe that this limitation is not unique to our study, but rather a limitation of essentially all NF training studies. Therefore, we have emphasized the need for future NF training research to challenge this assumption as noted above by the revision in the Conclusions section.

---

## [Decision Letter · Decision Letter 3]

9 Mar 2023

Neural and behavioral adaptations to frontal theta neurofeedback training: A proof of concept study

PONE-D-21-31557R3

Dear Dr. Kerick,

We’re pleased to inform you that your manuscript has been judged scientifically suitable for publication and will be formally accepted for publication once it meets all outstanding technical requirements.

Kind regards,

Thalia Fernandez, Ph.D.

Academic Editor

PLOS ONE

Additional Editor Comments (optional):

Reviewers' comments:

Reviewer's Responses to Questions

**Comments to the Author**

1. If the authors have adequately addressed your comments raised in a previous round of review and you feel that this manuscript is now acceptable for publication, you may indicate that here to bypass the “Comments to the Author” section, enter your conflict of interest statement in the “Confidential to Editor” section, and submit your "Accept" recommendation.

Reviewer #3: All comments have been addressed

2. Is the manuscript technically sound, and do the data support the conclusions?

Reviewer #3: Yes

3. Has the statistical analysis been performed appropriately and rigorously? 

Reviewer #3: Yes

4. Have the authors made all data underlying the findings in their manuscript fully available?

Reviewer #3: Yes

5. Is the manuscript presented in an intelligible fashion and written in standard English?

Reviewer #3: Yes

6. Review Comments to the Author

Reviewer #3: (No Response)

7. PLOS authors have the option to publish the peer review history of their article (what does this mean?). If published, this will include your full peer review and any attached files.

Reviewer #3: **Yes: **Mauricio González-López

---

## [Editor Report · Acceptance letter]

14 Mar 2023

PONE-D-21-31557R3 

Neural and behavioral adaptations to frontal theta neurofeedback training: A proof of concept study 

Dear Dr. Kerick:

I'm pleased to inform you that your manuscript has been deemed suitable for publication in PLOS ONE. Congratulations! Your manuscript is now with our production department. 

Kind regards, 

on behalf of

Dr. Thalia Fernandez 

Academic Editor

PLOS ONE